# Annual greenhouse gas budget for a bog ecosystem undergoing restoration by rewetting

Sung-Ching Lee[1], Andreas Christen[1], Andrew T. Black[2], Mark S. Johnson[3,4], Rachhpal S. Jassal[2], Rick Ketler[1], Zoran Nesic[1,2], Markus Merkens[5]

[1]Department of Geography / Atmospheric Science Program, The University of British Columbia, Vancouver, Canada
[2]Faculty of Land and Food Systems, The University of British Columbia, Vancouver, Canada
[3]Institute of Resources, Environment and Sustainability, The University of British Columbia, Vancouver, Canada
[4]Department of Earth, Ocean and Atmospheric Sciences, The University of British Columbia, Vancouver, Canada
[5]Parks, Planning and Environment Department, Metro Vancouver, Vancouver, Canada

*Correspondence to*: S.-C. Lee (sungching.lee@geog.ubc.ca)

**Abstract.** Many peatlands have been drained and harvested for peat mining, agriculture, and other purposes, which has turned them from carbon (C) sinks into C emitters. Rewetting of disturbed peatlands facilitates their ecological recovery, and may help them revert to carbon dioxide ($CO_2$) sinks. However, rewetting may also cause substantial emissions of the more potent greenhouse gas (GHG) methane ($CH_4$). Our knowledge on the exchange of $CO_2$ and $CH_4$ following rewetting during restoration of disturbed peatlands is currently limited. This study quantifies annual fluxes of $CO_2$ and $CH_4$ in a disturbed and rewetted area located in the Burns Bog Ecological Conservancy Area in Delta, BC, Canada. Burns Bog is recognized as the largest raised bog ecosystem on North America's West Coast. Burns Bog was substantially reduced in size and degraded by peat mining and agriculture. Since 2005, the bog has been declared a conservancy area, with restoration efforts focusing on rewetting disturbed ecosystems to recover *Sphagnum* and suppress fires. Using the eddy covariance (EC) technique, we measured year-round (16th June 2015 to 15th June 2016) turbulent fluxes of $CO_2$ and $CH_4$ from a tower platform in an area rewetted for the last 8 years. The study area, dominated by sedges and *Sphagnum*, experienced a varying water table position that ranged between 7.7 (inundation) and -26.5 cm from the surface during the study year. The annual $CO_2$ budget of the rewetted area was -179 ± 26.2 g $CO_2$-C m$^{-2}$ year$^{-1}$ ($CO_2$ sink) and the annual $CH_4$ budget was 17 ± 1.0 g $CH_4$-C m$^{-2}$ year$^{-1}$ ($CH_4$ source). Gross ecosystem productivity (GEP) exceeded ecosystem respiration ($R_e$) during summer months (June-August), causing a net $CO_2$ uptake. In summer, high $CH_4$ emissions (121 mg $CH_4$-C m$^{-2}$ day$^{-1}$) were measured. In winter (December-February), while roughly equal magnitudes of GEP and $R_e$ made the study area $CO_2$ neutral, very low $CH_4$ emissions (9 mg $CH_4$-C m$^{-2}$ day$^{-1}$) were observed. The key environmental factors controlling the seasonality of these exchanges were downwelling photosynthetically active radiation and 5-cm soil temperature. It appears that the high water table caused by ditch blocking suppressed $R_e$. With low temperatures in winter, $CH_4$ emission was more suppressed than $R_e$. Annual net GHG flux from $CO_2$ and $CH_4$ expressed in terms of $CO_2$ equivalents ($CO_2e$) during the study period totalled to -22 ± 103.1 g $CO_2e$ m$^{-2}$ year$^{-1}$ (net $CO_2e$ sink) and 1248 ± 147.6 g $CO_2e$ m$^{-2}$ year$^{-1}$ (net $CO_2e$ source) by using 100-year and

20-year global warming potential values, respectively. Consequently, the ecosystem was almost $CO_2e$ neutral during the study period expressed on a 100-year time horizon but was a significant $CO_2e$ source on a 20-year time horizon.

## 1 Introduction

Wetland ecosystems play a disproportionately large role in the global carbon (C) cycle compared to the surface area they occupy. Wetlands cover only 6% − 7% of the Earth's surface (Lehner and Döll, 2004; Mitsch et al., 2010) but C storage in wetlands has been estimated to be up to 450 Gt C or approximately 20% of the total C storage in the terrestrial biosphere (Bridgham et al., 2006; Lal, 2008; Wisniewski and Sampson, 2012). On the other hand, they emit significant quantities of methane ($CH_4$), a powerful greenhouse gas (GHG), which is responsible for 30% of all global $CH_4$emissions (Bergamaschi et al., 2007; Bloom et al., 2010; Ciais et al., 2013) due to anaerobic microbial decomposition (Aurela et al., 2001; Rinne et al., 2007). Peatlands are the most widespread of all wetland types in the world representing 50 to 70% of global wetlands (Roulet, 2000; Yu et al., 2010. Peatlands around the world sequester around 50 g $CO_2$-C $m^{-2}$ $year^{-1}$ (Roulet et al., 2007; Christensen et al., 2012; Humphreys et al., 2014; McVeigh et al., 2014; Peichl et al., 2014, Pelletier et al., 2015) and emit around 12 g $CH_4$-C $m^{-2}$ $year^{-1}$ (Abdalla et al., 2016; Brown et al., 2014; Jackowicz-Korczynski et al., 2010; Lai et al., 2014; Urbanova et al., 2013). Futhermore, it has been shown that it is crucial to include peatlands in the modelling and analysis of the global C cycle (Frolking et al., 2013; Kleinen et al., 2010; Wania et al., 2009).

Many peatlands have been harvested and continue to be disturbed by the extraction of peat for horticultural use and conversion to agriculture as well as other purposes. In the case of Burns Bog, peat was also used for fire bombs during World War II (Cowen, 2015). Generally, during harvesting, the surface vegetation is removed, and then wetlands are drained by a network of ditches (Price and Waddington, 2000; Waddington and Roulet, 2000). When no longer economical, many harvested peatlands are abandoned and kept at artificially low water tables due to the drainage ditches. This environmental condition limits the disturbed and abandoned peatlands ability to return to their prior state. Drainage results in increased oxidation in peat soils, which then can become a strong source of $CO_2$ (Langeveld et al., 1997; Petrescu et al., 2015; Tapio-Biström et al., 2012). Additionally, degraded peat increases the risk of peatland fires, which could consequently cause significant $CO_2$ emissions (Gaveau et al., 2014; Page et al., 2002; van der Werf et al., 2004). These consequences could be worse if nothing is done after the peat extraction. Therefore, and for reasons of conservation ecology (unique habitat), disturbed peatlands may be restored.

Restoration efforts typically rely on elevating the water table and managing vegetation. The water table depth and the amount of vegetation are the most important factors affecting land-atmosphere C exchange. Rewetting by ditch blocking can have an immediate impact on the C exchange between the peatland surface and the atmosphere (Limpens et al., 2008). Rewetting has strong direct and indirect effects on $CO_2$ and $CH_4$ fluxes. Raising the water level has been found to suppress the $CO_2$ efflux from the soil and result in an increase in net $CO_2$ uptake by native bog vegetation (Komulainen et al., 1999). $CH_4$ emissions from rewetted sections in a bog in Finland were three times higher than the release from the disturbed and dry

area (Tuittila et al., 2000). Another study found similar rates of $CH_4$ production in disturbed and restored wetlands in the southern United States (Schipper and Reddy, 1994). Re-vegetation of degraded peat leads to faster re-establishment of peat formation that can have significant effects on C exchange. However, the increased above- and below-ground biomass of plants and litter enhances organic matter oxidation, which raises $CO_2$ emissions (Finér and Laine, 1998; Minkkinen and Laine, 1998). In other studies, re-establishing the conditions permitting peat formation also initially increased $CH_4$ emission,

but the C exchange did not reach the level of seasonal emissions from pristine peatlands (Crill et al., 1992; Dise et al., 1993; Shannon and White, 1994).

     Very few studies provide continuous, year-round measurements to determine how restored and rewetted peatland ecosystems recover in terms of their productivity and GHG exchange. It remains unclear when, or even if, restored peatland ecosystems could show a similar magnitude of C fluxes as in pristine (undisturbed) peatland ecosystems. Furthermore, most

investigation focusing on GHG exchange of restored peatlands only measured $CO_2$ and/or $CH_4$ fluxes during short periods, e.g. the growing season. There are few studies that measured continuously and year-round fluxes (Anderson et al., 2016; Järveoja et al., 2016; Knox et al., 2015; Richards and Craft, 2015; Strack and Zuback, 2013), relying instead on sporadic, or repeating chamber measurements, which are difficult to upscale to annual totals.

     In this study, we a) quantified seasonal and annual $CO_2$ and $CH_4$ fluxes, using the eddy covariance (EC) technique, in a

disturbed ecosystem that is representative of areas subject to recent restoration efforts (ditch blocking for the last 8 years), b) identified key environmental controls and their effects on $CO_2$ and $CH_4$ fluxes, and c) quantified whether the study ecosystem is net source or sink of C and its net climate forcing at different time scales by considering GWPs of $CO_2$ and $CH_4$.

## 2 Study area

Burns Bog in Delta, BC, on Canada's Pacific Coast, is part of a remnant peatland ecosystem that is recognized as the largest raised bog ecosystem (2,042 ha) on North America's west coast. During the last century, it was significantly disturbed as a result of it being used for housing, peat mining and agriculture (MetroVancouver, 2007). The Burns Bog Ecological Conservancy Area (BBECA) was established in 2005 to conserve this large coastal raised bog and restore ecological integrity to the greatest extent possible. Christen et al. (2016) measured summertime $CO_2$ and $CH_4$ exchanges using

primarily chamber systems in several plots representative of disturbed areas of the BBECA, where some plots were rewetted and others were not. The study found substantial emissions of $CH_4$ primarily in recently rewetted plots, with highest emissions associated with high water tables. Nevertheless, a significant spatial and temporal variability was found between and within plots. In order to constrain these emission estimates, it was suggested to extend the year-round monitoring of $CO_2$ and $CH_4$ exchanges using EC technique to provide spatially more representative fluxes at a recently rewetted plot.

The current study site is located in a harvested, disturbed, and rewetted area in the centre of the BBECA (122°59'05.87"W, 49°07'47.20"N, WGS-84) with dimensions of 400 m by 250 m (Fig. 1). The field is surrounded by a

windbreak to the west and an abandoned (now blocked) drainage ditch to the north (see supplementary material, Fig. S1 and S2). The study area was harvested between 1957 and 1963 using the Atkins-Durbrow Hydropeat method to remove the peat (Heathwaite and Göttlich, 1993). In 2007, the study site was rewetted via ditch-blocking using dams built with plywood and using wooden stakes as bracing (Howie et al., 2009). Based on the weather data for 1981 to 2010 from the closest Environment Canada weather station, Vancouver International Airport, the average annual temperature was 10.4 ℃ and average annual precipitation was 1062 mm. Following rewetting, water table height (WTH) in the study area fluctuates between 30 cm above ground and 20 cm below ground over the year. In all years since rewetting started in 2007, water table positions were lower in late summer and early fall and high all winter and spring. WTH decreases steadily between June and September. In September and October, a water table rise due to the increase in precipitation and reduced evapotranspiration (ET) (Fig. 2) as a consequence of reduced available energy and senescence of sedges was observed, which is similar to water table observations in other temperate wetlands (Lafleur et al., 2005; Rydin et al., 2013) . The depth of peat at the study site is 5.83 m. A silty clay layer is located below the peat layer (Chestnutt, 2015). The plant communities in the study ecosystem are dominated by *Sphagnum spp.* and *Rhynchospora alba*. The average height of the vegetation during the growing season is about 0.3 m (Madrone Consultants Ltd., 1999). Plants are separated by shallow open water pools, some of them populated by algae developing. Birch trees are dispersed and appear to be growing on the remnants of baulks but none of them was taller 2 m. Sphagnum covers over 25% of the surface inside the study area (Hebda et al., 2000). The area of the open water ponds was estimated to be about 20% of the surface in summer by aerial photo.

## 3 Materials and methods

### 3.1 Climate measurements

Weather variables were continuously measured in order to determine climatic controls of $CO_2$ and $CH_4$ fluxes. Four components of radiation (shortwave/longwave, incoming and outgoing) were continuously measured by a four-component net radiometer (CNR1, Kipp and Zonen, Delft, Holland) on top of the tower. Two quantum sensors (LI-190, LI-COR Inc., Lincoln, NE, USA) measured incoming and outgoing photosynthetically active radiation (PAR). Precipitation was measured with an unheated tipping bucket rain gauge (TR-525M, Texas Electronics, Dallas, TX, USA) at 1 m height, 10 m north of the tower. Air temperature ($T_a$) and relative humidity (RH, HMP-35 A, Vaisala, Finland) were measured at the heights of 2.0 m and 0.3 m, and soil thermocouples (type T) were recording soil/water temperatures at the depths of 0.05, 0.10 and 0.50 m ($T_{s,5cm}$, $T_{s,10cm}$, and $T_{s,50cm}$). A pressure transducer (CS400, CSI) was installed on July 28[th] 2015 in an observation well west of the tower to continuously measure WTH for the remainder of the study period. A soil volumetric water content ($\theta_w$) sensor (CS616, CSI) was inserted vertically to measure integrated $\theta_w$ from the surface to a depth of 0.30 m.

## 3.2 Eddy-covariance measurements

Over the entire annual study period, from 16[th] June 2015 to 15[th] June 2016, a long-term eddy covariance system (EC-1) was operated on a floating scaffold tower (Fig. 1) at a height of 1.8 m (facing south). The EC-1 system consisted of an ultrasonic anemometer-thermometer (CSAT-3, Campbell Scientific Inc. (CSI)) and an open-path $CO_2$/$H_2O$ infrared gas analyzer (IRGA, LI-7500, LI-COR Inc.). The path separation between CSAT-3 and LI-7500 was 5 cm. The CSAT-3 measured the longitudinal, transverse and vertical components of the wind vector and sonic temperature and output data at 10 Hz. The IRGA measured water vapor density ($\rho_v$) and $CO_2$ density ($\rho_c$) at 10 Hz. The 10-Hz data from both instruments were sampled on a data logger (CR1000, CSI) and processed fluxes of $CO_2$ (NEE) were calculated in post-processing of 30-min data blocks following the procedures documented in Crawford et al. (2013) .

An additional, independent EC system (EC-2) was added on June 10[th] 2015 to measure $CH_4$ fluxes. The EC-2 system was also located at a height of 1.8 m, 1.8 m to the west of EC-1, and faced south (Fig. 1). EC-2 consisted of a similar ultrasonic anemometer-thermometer (CSAT-3, CSI, 20 Hz), an enclosed-path $H_2O$/$CO_2$ IRGA (LI-7200, LI-COR Inc., 20 Hz) and an open-path gas analyzer to measure the partial density of $CH_4$ ($\rho_m$) (LI-7700, LI-COR Inc., 20 Hz). The northward-separation of LI-7200 was 20 cm. The northward-separation of LI-7700 was 40 cm and eastward-separation of LI-7700 was 20 cm. Data from EC-2 were collected by an analyzer interface unit (LI-7550, LI-COR Inc.) and processed on-site. Fluxes of $CH_4$ ($F_m$) were processed in advanced mode using EddyPro® (V6.1.0, LI-COR Inc.) with a missing sample allowance of 30%. $F_m$ data were quality checked using the flagging system proposed by Mauder and Foken (2004).

## 3.3 Gap filling algorithms

Some gaps in climate and flux measurements are unavoidable due to challenging weather and low-light situations (the station was solar powered), and need to be filled in for estimating seasonal and annual fluxes. Gaps in the climate data (<1% of the year) were filled using measurements at nearby climate stations. Small gaps (<60 minutes) of missing $CO_2$, $H_2O$, and $CH_4$ fluxes were filled by linear interpolation. Longer gaps in $H_2O$ fluxes were filled with the online tool developed by the Max Planck Institute for Biogeochemistry in Jena, Germany. This tool uses the look-up table method documented in Falge et al. (2001) and Reichstein et al. (2005). Longer gaps in $CO_2$ and $CH_4$ fluxes were filled using empirical relationships between $CO_2$ or $CH_4$ fluxes and environmental variables. Two-year (from July 2014 to June 2016) of measurements of $CO_2$ fluxes were used for modelling $R_e$ and GEP to achieve better statistical relationships. Since there were two EC systems running with redundant fluxes of $CO_2$, the sensitivity of different combinations of data (EC-1 vs. EC-2 or using an average of the two) has been explored in Lee et al. (2016). For the data presented in this study, $CO_2$ fluxes, $H$, $LE$ from EC-1 and $CH_4$ fluxes from EC-2 were used. Valid data from EC-1 were obtained for 59% of the year (after quality control). Valid data from EC-2, which were restricted by power availability, were 32% of the year (after quality control). Data availability was the lowest in winter (38%/4% in winter, 71%/6% in spring, 67%/70% in summer, 60%/51% in fall, for EC-1/EC-2, respectively).

In this study, net fluxes of $CO_2$ and $CH_4$ toward the ecosystem surface are negative and net fluxes from the ecosystem surface to the atmosphere are positive. Therefore, negative NEE and $F_m$ represent net $CO_2$ and $CH_4$ uptake, respectively.

### 3.3.1 Gap filling of $CO_2$ flux data

For gaps longer than 2 hours in $CO_2$ fluxes, the $CO_2$ flux (i.e., net ecosystem exchange, NEE) was modelled as the difference between ecosystem respiration ($R_e$) and gross ecosystem productivity (GEP) i.e. $NEE = R_e − GEP$. Nocturnal NEE values were $R_e$ as there is no photosynthesis at night.

$R_e$ was modelled based on soil temperature at the 5-cm depth ($T_{s,5cm}$) using a logistic fit (Neter et al., 1988):

$$R_e = \frac{1}{r_1 r_2^{T_{s,5cm}} + r_3} \tag{1}$$

A comparable logistic function was proposed and used by FLUXNET Canada (Barr et al., 2002; Kljun et al., 2006). In this study, we used this logistic model available in IDL (version 8.5.1, Exelis Visual Information Solutions, Boulder, Colorado). $r_1, r_2,$ and $r_3$ are empirical parameters; $r_1$ controls the slope of exponential phase; $r_2$ determines where the transitional phase starts; and $r_3$ determines the height of plateau phase. For each day of the year, the parameters $r_1, r_2,$ and $r_3$ for $R_e$ were determined independently using a moving $\pm$ 60-day window centered on that day based on all measured nighttime data from 2014 to 2016 when friction velocity was higher than 0.08 m s$^{-1}$. Lee (2016) determined the effect of using different window sizes (60, 90, 120 and full year) on the annual modelled and gap-filled $R_e$ and showed that a moving window size of 120 days was least sensitive to errors while still allowing for seasonal changes. However, sensitivity of choosing different
window sizes on gap filled $R_e$ was small, varying the annual value between 226 and 245 g C m$^{-2}$ year$^{-1}$.

GEP was first partitioned from measured daytime NEE using modelled $R_e$. Any missing GEP data were then modelled using the photosynthetic light-response curves (Ö gren and Evans, 1993) based on photosynthetic photon flux density (PPFD in µmol m$^{-2}$ s$^{-1}$):

$$GEP = \frac{MQY \cdot PPFD + P_M - ((MQY \cdot PPFD + P_M)^2 - 4 \cdot C_v \cdot MQY \cdot PPFD \cdot P_M)^{0.5}}{2 \cdot C_v} \tag{2}$$

Maximum photosynthetic rate at light saturation ($P_M$) and maximum quantum yield ($MQY$) are fitted parameters with GEP estimated as measured daytime NEE minus daytime $R_e$ calculated using Eq. 1. Convexity ($C_v$) was fixed at 0.7 (Farquhar et al., 1980). For each day of the year, the time-varying parameters $MQY$ and $P_M$ were determined independently using a
moving $\pm$ 45-day window centered on that day using all data from 2014 to 2016 when friction velocity was higher than 0.08 m s$^{-1}$. The sensitivity of window size on gap filled GEP was small, resulting in annual value to vary between 385 and 415 g C m$^{-2}$ year$^{-1}$.

### 3.3.2 Gap filling of CH$_4$ flux data

CH$_4$ fluxes with quality flags 0 and 1 according to Mauder and Foken (2004) were plotted against all relevant variables
including NEE, $WTH$, $\theta_w$, $T_a$, $T_{s,5cm}$, $T_{s,10cm}$, and $T_{s,50cm}$. The highest correlation between a single variable and the CH$_4$ flux
was found for soil temperature using an exponential relationship (Fig. S3). Of the soil temperatures measured at three
different depths, $T_{s,10cm}$ explained the highest proportion of the variance in CH$_4$ flux (Table S1). Therefore, $T_{s,10cm}$ was used
to build an initial model and a logarithmic transformation of the CH$_4$ fluxes was applied to remove the heteroscedasticity and
permit the use of a linear regression model. Then the residual analysis was applied to explore whether the variance in the
residual could be explained by other controls. The residual was defined as the ratio of the measured CH$_4$ fluxes to the
modelled CH$_4$ fluxes from the initial model. Based on the residual analysis, the main contributor to the residual, $WTH$,
explained 7% of the variance (Table S2). Additionally, there was a hysteresis relationship between CH$_4$ flux and $WTH$ (Fig.
S4). In order to have a more robust gap filling model, $T_{s,10cm}$ and $WTH$ were used to fill the gaps in CH$_4$ fluxes. We used a
combination of an exponential temperature response function and a linear $WTH$ function as follows:

$$F_m = (aWTH + b)e^{cT_{s,10cm}} \tag{3}$$

where $a$, $b$, and $c$ are time-varying empirical parameters. The three parameters were fitted separately for each day, using a
moving window of $\pm105$ days using all data from the study period when friction velocity was greater than 0.08 m s$^{-1}$. Overall,
76% of the variance of the CH$_4$ fluxes was explained by $T_{s,10cm}$ and $WTH$. The combination of soil temperature and $WTH$ has
also been shown to explain a large proportion of the observed variances in CH$_4$ fluxes in peatlands in other studies (Brown et
al., 2014; Goodrich et al., 2015).

### 3.3.3 Error estimates

The uncertainty associated with annual estimates of NEE, GEP, $R_e$ and CH$_4$ fluxes resulting from gap filling and due to
different window sizes was quantified as follows: First, in the annual dataset of half-hourly fluxes random gaps were inserted
using Monte Carlo simulation (Griffis et al., 2003; Krishnan et al., 2006; Paul-Limoges et al., 2015); The maximum number
of gaps were set to 40 and the maximum length was set to 10 days resulting in total gaps of on average 28% of the year (and
up to 40% of the year). The Monte Carlo simulation was run 500 times and the 95% confidence intervals were used to
calculate the uncertainty of the annual sums.

Secondly, the uncertainty associated with choosing different window sizes for the derivation of the relationships in the
gap-filling (see Section 3.3.1 and 3.3.2) was estimated from a range of annual values obtained using window sizes of 30, 45,
60, 75, 90, 120, 150, 180, and 365 days for GEP, $R_e$, and NEE; the same selections of window sizes with three additions (210,
240, and 270 days) were applied for calculating the uncertainty of the annual CH$_4$ budget. The overall uncertainty in the

annual estimates of NEE, GEP, $R_e$ and $CH_4$ fluxes was then obtained by taking the square root of the sum of squares of the error from the gap filling (Monte Carlo simulation) and the uncertainty of the estimates due to different window sizes.

## 3.4 Calculating $CO_2$e

The combined effect of all long-lived greenhouse gases was compared for $CO_2$ and $CH_4$ by converting the molar fluxes of $CO_2$ and $CH_4$ into time-integrated radiative forcing (i.e. global warming potential, GWP) expressed on a mass basis in terms of $CO_2$ equivalents (g $CO_2$e $m^{-2}$ $s^{-1}$) as follows:

$$CO_2\text{e (g)} = m_{CO_2}F_c + GWP_{CH_4}m_{CH_4}F_m \tag{4}$$

where $GWP_{CH_4}$ is the mass-based GWP for the $CH_4$ (g $g^{-1}$), $m_{CO_2}$ is the molecular mass of $CO_2$ (44.01 g $mol^{-1}$), and $m_{CH_4}$ is the molecular mass of $CH_4$ (16.04 g $mol^{-1}$). In this study, a 100-year GWP of $CH_4$ of 28, and 20-year GWP of $CH_4$ of 84, were used respectively (IPCC, 2014). $N_2O$ fluxes have been neglected in this study because previous chamber-based measurements during the growing season found no significant emissions or uptake of $N_2O$ in all study plots in the BBECA (Christen et al., 2016).

## 4 Results and Discussion

### 4.1 Weather

During the study period (June 16[th] 2015 to June 15[th] 2016), the site experienced an annual average $T_a$ (2 m height) of 11.3 °C. Mean monthly $T_a$ ranged between 4.4 (Jan 2016) and 19.3 °C (Jul 2015). The study site received a total annual precipitation of 1062 mm, of which 16% (174 mm) fell during the warm half year (Apr-Sep) and 84% (888 mm) during the cold half year (Oct-Mar) (Fig. 2). There was no lasting snow cover during the study year. However, the surface was frozen over ten days in January 2016, with an ice thickness of up to 5 cm.

Winds at this site were often influenced by a sea-land breeze circulation. Under sea-breeze situations, wind mainly came from the south (40% of all cases). Sometimes, however, the sea-land breeze blew from the west, primarily between 17:00 and 19:00 PST. The wind direction on average turned to east during the nighttime (land-breeze), and generally at night, the winds were weaker.

### 4.2 Surface conditions

#### 4.2.1 Turbulent flux footprints

Cumulative turbulent source areas were calculated using the analytical turbulent source area (turbulent footprint) model (Kormann and Meixner, 2001) following the procedure outlined in Christen et al. (2011). The 80% contour line (enclosing

80% of the cumulative probability for a unit source) was entirely inside the field in spring and summer. It reached beyond the ditches at the north side in fall and winter. Unstable conditions during daytime allowed for a more constrained footprint surrounding the tower. Stable conditions at night led to larger footprints, primarily from East. The cumulative footprint for each of the four seasons for the EC-1 overlaid on the satellite image of the site are documented in Fig. S1 (supplementary material).

### 4.2.2 Vegetation cover and water table changes

Mosses and white beak sedge (the common name of *Rhynchospora alba*) started to grow in March and grasses grew up to a maximum of 0.3 m height in summer. In summer, vegetation covered almost the entire study area of the surface, including ponds (some with algae), so the surface was less patchy in summer compared to other seasons, when standing water ponds were intermixed with vegetation in fall, winter and spring (see supplementary material, Fig. S2).

Winter was the wettest season when WTH was mostly above the bare soil (reference surface). The highest water table position was 7.7 cm above the reference surface in December. In the dry season, the water table position dropped to 26.5 cm beneath the bog surface in August. The WTH decreased in spring, and dry hummocks could be seen from April to September. The water table started to rise above the surface after receiving the fall precipitation. The study site was flooded in winter during the study year.

### 4.3 CO$_2$ exchange

### 4.3.1 Annual, seasonal and monthly NEE, $R_e$ and GEP

Overall, the study area was a CO$_2$ sink in spring (MAM, -1.10 g C m$^{-2}$ day$^{-1}$) and in summer (JJA, -0.82 g C m$^{-2}$ day$^{-1}$). Net CO$_2$ fluxes were near zero in fall (SON, +0.03 g C m$^{-2}$ day$^{-1}$) and winter (DJF, -0.07 g C m$^{-2}$ day$^{-1}$). Over the entire year, the annual CO$_2$-C budget (i.e., NEE) was -179 $\pm$ 26.2 g C m$^{-2}$ yr$^{-1}$. Almost in each month of the calendar year, the site was a weak sink for CO$_2$ except in October, November and December (Fig. 3, Table 1). Monthly net fluxes of CO$_2$ (NEE) ranged from +1.77 g C m$^{-2}$ month$^{-1}$ in November 2015 to -56.20 g C m$^{-2}$ month$^{-1}$ in May 2016.

The annual $R_e$ and GEP during the study year were 236 $\pm$ 16.4 and 415 $\pm$ 28.8 g C m$^{-2}$ yr$^{-1}$, respectively. The relative changes in $R_e$ and GEP were closely linked to the seasonality of the plant phenology. Based on GEP trends, we can divide the study period into three segments, 'winter' (Oct-Mar), 'early growing season' (Apr-Jun), and 'late growing season (Jul-Sep). The rising temperature triggered growth in the early growing season (GEP = 59.73 g C m$^{-2}$ month$^{-1}$), while the later growing season had limited growth (GEP = 25.08 g C m$^{-2}$ month$^{-1}$). Winter had lowest productivity (GEP = 7.58 g C m$^{-2}$ month$^{-1}$) (Table 1). Compared to a large seasonal amplitude in monthly GEP, $R_e$ showed less variability over the year. The highest rate of increase in the magnitude of NEE and the highest magnitude of NEE both occurred early in the growing season (Fig. 3). This was caused by the onset of $R_e$ being delayed compared to GEP, resulting in the greatest imbalance between respiratory and assimilatory fluxes in May.

Table 2 compares annual NEE, $R_e$ and GEP at the study site to Fluxnet sites over other land covers in the same region that experienced similar climate forcings, although from different years. An unmanaged grassland site 15 km to the west of the study area in the Fraser River Delta (Westham Island, Delta, BC, Crawford et al., 2013) had about 1.3 times higher NEE than this rewetted area. Annual $R_e$ and GEP values at this grassland site were higher than the study site by a factor of 5.2 and 3.5. A mature 55-year-old Douglas-fir forest on Vancouver Island (200 km NW of the study area; Krishnan et al., 2009) showed an NEE of 1.8 times higher than the study area. The $R_e$ and GEP were even higher by factors of 7.8 and 5.2, respectively. A young forest plantation (Buckley Bay, 150 km W of the study area; Krishnan et al., 2009), which was a weak C source, had $R_e$ and GEP of six- and three-fold higher than the study site, respectively. Compared to these other sites under similar climatic conditions, the rewetted area of the bog was not an ecosystem of high productivity but one with considerably limited $R_e$ that permits more efficient $CO_2$ sequestration (-NEE is 43 % of GEP, as opposed to 15% for the unmanaged grassland site and mature forest).

The annual NEE in this study was more negative than in the majority of previously reported NEE values for undisturbed temperate peatlands, which were weak sinks, typically in the range of -50 g C m$^{-2}$ year$^{-1}$ (Christensen et al., 2012; Humphreys et al., 2014; Matthias et al., 2014; McVeigh et al., 2014; Pelletier et al., 2015; Roulet et al., 2007). Values that are comparable to the current restored wetland were reported in five pristine temperate wetlands: $-248$ g C m$^{-2}$ year$^{-1}$ (Lafleur et al., 2001), $-234$ g C m$^{-2}$ year$^{-1}$ (Campbell et al., 2014), -210 g C m$^{-2}$ year$^{-1}$ (Fortuniak et al., 2017), $-189$ g C m$^{-2}$ year$^{-1}$ (Flanagan and Syed, 2011), and $-103$ g C m$^{-2}$ year$^{-1}$ (Lund et al., 2010). The few datasets in the literature for NEE of restored wetlands showed a wide range of values. Some were $CO_2$ sources, with NEE ranging from +103 g C m$^{-2}$ year$^{-1}$ to +142 g C m$^{-2}$ year$^{-1}$ (Järveoja et al., 2016; Richards and Craft, 2015; Strack and Zuback, 2013). Other measurements, however, showed that restored wetlands were sinks, all of them stronger than in this study, with NEE values ranging from -446 g C m$^{-2}$ year$^{-1}$ to -270 g C m$^{-2}$ year$^{-1}$ (Badiou et al., 2011; Hendriks et al., 2007; Herbst et al., 2013; Knox et al., 2015). In this study, values of $R_e$ and GEP were lower than those found for a restored wetland at a comparable latitude in the central Netherlands with slightly lower annual temperature and precipitation (Hendriks et al., 2007). $R_e$ and GEP in this study area were also lower than values for most pristine peatlands at comparable latitudes (Helfter et al., 2015; Levy and Gray, 2015). Comparably low $R_e$ and GEP were reported from the Mer Bleue boreal raised bog (Lafleur et al., 2001; Moore, 2002) and from an Atlantic blanket bog (McVeigh et al., 2014; Sottocornola and Kiely, 2010), both of which had a lower mean annual temperature than Burns Bog.

It is important to estimate dissolved organic carbon (DOC) export to determine a more complete ecosystem C budget. DOC lost from restored and pristine peatlands have been found typically to range from 3.4 to 16.1 g C m$^{-2}$ year$^{-1}$ (Hendriks et al., 2007; Koehler et al., 2011; Roulet et al., 2007; Waddington et al., 2010), although, Chu et al. (2014) reported a net DOC import for a marsh of $23 \pm 13$ g C m$^{-2}$ year$^{-1}$. Estimation of DOC fluxes was based on regular (approx. monthly) water samples collected at 5 locations within the flux tower footprint. Water samples were analyzed for DOC concentrations using a TOC analyzer (Model TOC-VCSH, Shimadzu Scientific, Kyoto, Japan). Lateral water export was estimated as the residual

of the water balance. D'Acunha et al. (2016) estimated DOC export for the current study area for Jan – Dec 2016 to be 22.4 g C m$^{-2}$ year$^{-1}$ (15% of annual NEE).

### 4.3.2 Diurnal variability in CO$_2$ fluxes

The seasonally-changing diurnal course of gap-filled NEE with isopleths over time of day and year is shown in Fig. 4. The daily maximum in GEP changed with season resulting in the high magnitude of NEE during midday between May and July (~ -3.5 µmol m$^{-2}$ s$^{-1}$) with the highest magnitude of NEE occurring in May. Nighttime NEE, i.e., $R_e$, showed relatively small variation with season, and on average was ≤1 µmol m$^{-2}$ s$^{-1}$ for most of the study period. The rapid decrease in monthly $R_e$ from May to June (Table 1) was caused by low $R_e$ in early morning or at nightfall in June.

**4.3.3 Ecosystem respiration**

Figure 5 shows the relationship between nighttime $R_e$ and $T_{s,5cm}$ using the data for the entire study period. $R_e$ increased with increasing $T_{s,5cm}$ as expected, and annually followed a logistic curve rather than an exponential relationship. $R_e$ response curves were also calculated every two months (see supplementary material, Fig. S5). $R_e$ showed different curves depending on season. In winter, $R_e$ varied little with $T_{s,5cm}$ and was close to zero. From February to May, the relationship became closer

to logistic. In June and July, due to general warm condition (>15°C), $R_e$ remained nearly constant at ~1 µmol m$^{-2}$ s$^{-1}$ (the fitted curve stayed in the plateau phase). The study area had the highest $R_e$ in these two months. In fall, $R_e$ curves were closer to an exponential relationship, which could be due in part to leaf senescence (Shurpali et al., 2008). Decomposition of dead plant organic matter on the soil surface may have caused a higher $R_e$ in fall compared to spring and winter at the same $T_{s,5cm}$. Another factor could be the WTH, which in fall was not high enough to suppress $R_e$ as it did in winter (Juszczak et al., 2013).

The differences between March and September $R_e$ at the same $T_{s,5cm}$ were up to 0.4 µmol m$^{-2}$ s$^{-1}$.

    Two other controls on $R_e$ explored were air temperature ($T_a$) and WTH. The role of WTH was described above and $T_a$ had a similar impact on $R_e$ as $T_{s,5cm}$ when $T_a < 16$°C, but for warmer temperatures, $T_a$ did not correlate with $R_e$. The explanation for this is that heterotrophic component of $R_e$ depends on $T_s$, not the rapidly changing $T_a$ (Davidson et al., 2002; Edwards, 1975; Lloyd and Taylor, 1994).

It is widely reported that in most terrestrial ecosystems, the activity of soil microbes is also governed by soil moisture status, having little activity when the soil is excessively dry or excessively wet. Accordingly, and like other wetlands, $R_e$ was small when the water table was above the surface because this situation suppressed aerobic decomposition of peat (Rochefort et al., 2002; Weltzin et al., 2000). When the water table was below surface, $R_e$ increased to near 1 µmol m$^{-2}$ s$^{-1}$ and became stable no matter how low the water table position was. This relationship was also found in many other peatlands (Bridgham

et al., 2006; Ellis et al., 2009; Strack et al., 2006). There was no obvious relationship between $\theta_w$ (integrated from 0-30 cm depth) and $R_e$. $R_e$ slightly decreased from 1.0 to 0.6 µmol m$^{-2}$ s$^{-1}$ when $\theta_w$ increased from 84% to 88%. Other than this range, $\theta_w$ had no more impact on $R_e$.

### 4.3.4 Gross ecosystem productivity

Figure 6 shows the average light response curve, with half-hourly GEP as a function of PPFD. Due to different phenology over the year and the changes in solar altitude, light response curves were also calculated every two months (see supplementary material, Fig. S6). GEP reached a maximum in May with 92.63 g C m$^{-2}$ month$^{-1}$, and a minimum of 2.79 g C m$^{-2}$ month$^{-1}$ in December (Fig. 3, Table 1). GEP at light saturation reached roughly 5.09 μmol m$^{-2}$ s$^{-1}$ in summer, and remained below 2.49 μmol m$^{-2}$ s$^{-1}$ in winter, due to reduced leaf area, flooding, and lower temperatures. From March to May, GEP increased much more rapidly than $R_e$. In fall, GEP decreased faster than $R_e$. The magnitude of $R_e$ already was close to GEP in the late August to make the study area become $CO_2$ neutral in late summer.

Other possible controls on GEP explored were WTH and $T_a$. We found that WTH was not a control on GEP ($R^2 = 0.08$) in the current study as the study area remained fairly wet throughout the year. Furthermore, the effect of $T_a$ on GEP was less and limited to a smaller temperature range, compared to $T_s$..

### 4.4 CH$_4$ exchange

### 4.4.1 Annual and seasonal CH$_4$ budgets

Overall, the study area was a source of $CH_4$ in each of the twelve months (Table 1). The annual $CH_4$-C budget was $17 \pm 1.0$ g $CH_4$-C m$^{-2}$ yr$^{-1}$. $CH_4$ emissions were close to zero in winter (5.2 mg $CH_4$-C m$^{-2}$ day$^{-1}$). Seasonally, it was a weaker $CH_4$ source in fall (31.3 mg $CH_4$-C m$^{-2}$ day$^{-1}$) and spring (36.4 mg $CH_4$-C m$^{-2}$ day$^{-1}$), and then became a much larger source in summer (126.0 mg $CH_4$-C m$^{-2}$ day$^{-1}$). Monthly emissions of $CH_4$ ranged from 93 (January) to 4371 (July) mg $CH_4$-C m$^{-2}$ month$^{-1}$. The rising $T_a$ did not trigger $CH_4$ production immediately, and $CH_4$ fluxes remained low in April and May. But once the subsurface and water became warm enough, $CH_4$ emissions increased from to 1.4 to 2.7 g $CH_4$-C m$^{-2}$ month$^{-1}$ in June (Table 1). $CH_4$ emissions reached the peak in July (4.4 g $CH_4$-C m$^{-2}$ month$^{-1}$) and held similar magnitude (3.8 g $CH_4$-C m$^{-2}$ month$^{-1}$) in August even though the $T_a$ had dropped. Although it has been suggested that in some peatlands, WTH acts as a main control on $CH_4$ fluxes (Drösler et al., 2008; Knorr et al., 2009; Romanowicz et al., 1995; Roulet et al., 1993; Windsor et al., 1992), it has also been found that $CH_4$ emissions from wet soils (where the water table fluctuates within a small range near the surface) are highly dependent on $T_s$ because the oxidation in a shallow top soil is negligible (Jackowicz-Korczynski et al., 2010; Long et al., 2010; Olson et al., 2013; Rinne et al., 2007; Song et al., 2009). In our study, $CH_4$ emissions in the summer months were relative high even when the water table dropped to around 20 cm below the surface, likely because the peat maintained anaerobic conditions above the water table (as discussed in Hendriks et al., 2007). In addition, one needs to consider the transport pathways for $CH_4$ which may help explain the higher $CH_4$ fluxes in summer. First, the presence of sedges created an effective additional diffusion pathway for $CH_4$ through the plants' aerenchyma (Herbst et al., 2011; Treat et al., 2007). Second, a high water table especially when it rises above the soil surface increases the diffusion resistance to $CH_4$ transport (Brown et al., 2014; Walter and Heimann, 2000).

The annual CH$_4$ flux in this study area was lower than CH$_4$ fluxes reported for other restored wetlands (Anderson et al., 2016; Hendriks et al., 2007; Knox et al., 2015; Mitsch et al., 2010). Despite the study area being flooded for most of the study year, CH$_4$ emissions were closer to fluxes measured over drained peatlands (Kroon et al., 2010; Schrier-Uijl et al., 2010). Only Herbst et al. (2013) reported an annual CH$_4$ flux from a restored wetland in Denmark that was lower than in this study (9 to 13 g CH$_4$-C m$^{-2}$ year$^{-1}$). Our annual CH$_4$ flux at 17 ± 1.0 g CH$_4$-C m$^{-2}$ year$^{-1}$ was comparable to an average natural temperate wetland CH$_4$ flux, which is typically around 15 g CH$_4$-C m$^{-2}$ year$^{-1}$ (Abdalla et al., 2016; Fortuniak et al., 2017; Nicolini et al., 2013; Turetsky et al., 2014). The CH$_4$ fluxes from a number of temperate and tropical pristine wetlands exceeded the CH$_4$ fluxes reported in this study, including emissions from marshes in the Southwestern US (130 g CH$_4$-C m$^{-2}$ year$^{-1}$, Whiting & Chanton, 2001), tropical wetlands in Costa Rica (82 g CH$_4$-C m$^{-2}$ year$^{-1}$, Nahlik & Mitsch, 2010), marshes in the Midwestern US (50 g CH$_4$-C m$^{-2}$ year$^{-1}$, Koh et al., 2009) ), all three studies based on chamber measurements, and an ombrotrophic bog in New Zealand (29 and 21 g CH$_4$-C m$^{-2}$ year$^{-1}$ based on EC measurements, Goodrich et al., 2015). However, all these studies were conducted using chambers and the sampling frequency was at most once per month.

### 4.4.2 Diurnal variability in CH$_4$ fluxes

The ensemble-averaged diurnal courses of the CH$_4$ fluxes measured by the EC-2 system are shown in Fig. 7 during the summer months due to the lack of missing wintertime data caused by power restriction. Surprisingly, there was only a small diurnal variation observed for CH$_4$ fluxes in the summer months, while larger diurnal variations have been found in other studies (Juutinen et al., 2004; Long et al., 2010; Sun et al., 2013; Wang and Han, 2005). In the current study area, with changes in WTH and vegetation growth occurring during the year, there were likely several processes affecting CH$_4$ transport, which masked the diurnal pattern of CH$_4$ fluxes. Furthermore, $T_{s,5cm}$ appeared to be the main environmental control on CH$_4$ fluxes in this study but did not have as strong effect on CH$_4$ emissions as found in previous studies. Thus CH$_4$ was continuously emitted at a similar rate during daytime and nighttime. From January to March and October to December, the winter half-year, the study site had constant CH$_4$ emissions of less than 50 nmol m$^{-2}$ s$^{-1}$, and almost no diurnal variation was observed. July had the greatest CH$_4$ emissions, and the highest magnitude (>150 nmol m$^{-2}$ s$^{-1}$) appeared in the evening (3 pm to 9 pm). This corresponded to the lagged effect of soil temperature and may be partly due to convective turbulent mixing caused by cooling during the evening (Godwin et al., 2013).

### 4.5 CO$_2$e balance

Figure 8a and 8b show CO$_2$ and CH$_4$ fluxes expressed in terms of CO$_2$e using 100-year and 20-year GWPs, respectively. Considering fluxes of both GHGs together, this rewetted area was annually near to CO$_2$e neutral at 100-year scale with a net uptake by CO$_2$ (-656 g CO$_2$e m$^{-2}$ year$^{-1}$) balanced by CH$_4$ emissions (634 g CO$_2$e m$^{-2}$ year$^{-1}$). On shorter time horizon of 20 years, the study area represented a significant C source in CO$_2$e terms as the net uptake of CO$_2$ (-656 g CO$_2$e m$^{-2}$ year$^{-1}$) was one-third that of CH$_4$ emissions (1904 g CO$_2$e m$^{-2}$ year$^{-1}$). In late spring and early summer, the early onset of CO$_2$ sequestration in May and the time lag in CH$_4$ fluxes combined to represent a negative net GHG forcing, no matter which

GWP time horizon was considered. The quick drop in $CO_2$ sequestration in August and September allowed the highest net GHG forcing to be observed at both time horizons in late summer. In short, the critical time period for both, $CO_2$ and $CH_4$ fluxes in terms of $CO_2e$, was the growing season when magnitude of fluxes changed differently across the growing season. The results show that measurements made during a part of the growing season are not necessarily representative for the entire growing season or the year; a short-term campaign can be a good way to identify important site processes but the determination of the annual budget requires reliable annual measurements.

Using GWP to classify a study area as a net GHG source or sink is useful; however, the appropriateness of this method in computing the actual radiative forcing has been questioned and alternative models have been proposed (Frolking and Roulet, 2007; Fuglestvedt et al., 2000; Neubauer and Megonigal, 2015; Petrescu et al., 2015; Smith and Wigley, 2000).

## 5 Conclusions

The study area, a rewetted plot in the BBECA undergoing ecological restoration, was a net $CO_2$ sink over the study period (-179 g ± 26.2 $CO_2$-C $m^{-2}$ $year^{-1}$). The study area was not a highly productive ecosystem (annual GEP = 415 ± 28.8 g $CO_2$-C $m^{-2}$ $year^{-1}$) but exhibited low $R_e$ (annual $R_e$ = 236 ± 16.4 g $CO_2$-C $m^{-2}$ $year^{-1}$), likely due to oxygen limitations. The annual $CO_2$ fluxes reported here from a restored and rewetted peatland are comparable with data reported from pristine temperate peatlands in temperate mid latitudes (Alm et al., 1997; Lafleur et al., 2001; Pihlatie et al., 2010; Shurpali et al., 1995). The study area sequestered less $CO_2$ than the few other restored wetlands reported in the literature (Anderson et al., 2016; Järveoja et al., 2016; Knox et al., 2015; Richards and Craft, 2015; Strack and Zuback, 2013). The major controls on $CO_2$ fluxes were PAR irradiance and $T_{s,5cm}$. The magnitude of PAR strongly controlled GEP, and the $T_{s,5cm}$ regulated $R_e$. WTH also had influence on $R_e$ especially when the ecosystem was flooded.

The annual $CH_4$ emission was 17 ± 1.0 g $CH_4$-C $m^{-2}$ $year^{-1}$, which is lower than values reported for other restored wetlands (Anderson et al., 2016; Knox et al., 2015). $CH_4$ emissions in the summer were 60 times stronger than in the winter. The ditch blocking resulted in anaerobic conditions with the water table being within 30 cm of the surface throughout the year. Effects of changing WTH on $CH_4$ fluxes at the study area were not clearly apparent. $T_{s,10cm}$ and WTH explained $CH_4$ fluxes best ($R^2$ = 0.76).

In terms of the C balance (excluding DOC fluxes), our results suggest that our study area in BBECA was a net C sink (-163 ± 26.2 g C $m^{-2}$ $year^{-1}$) during the 8th year following rewetting. Combining $CO_2$, $CH_4$ and DOC fluxes resulted in a net C balance of -141 ± 26.2 g C $m^{-2}$ $year^{-1}$. These results are consistent with those of several disturbed peatlands that have become a net annual C sink after following restoration by rewetting (Karki et al., 2016; Schrier-Uijl et al., 2014; Wilson et al., 2013). In terms of net climate forcing of the system related to $CO_2$ and $CH_4$ fluxes expressed by GWPs, our results show that the ecosystem was almost $CO_2e$ neutral ($CO_2e$ (g) = -22 ± 103.1 g $CO_2e$ $m^{-2}$ $year^{-1}$) over a 100-year time horizon.

**Acknowledgements**

This research was primarily funded through research contracts between Metro Vancouver and UBC (PI: Christen). Selected equipment was supported by the Canada Foundation for Innovation (Christen, Johnson) and NSERC RTI (Christen).
Financial support through scholarships and training were provided by UBC Faculty of Graduate and Postdoctoral Studies and UBC Geography. We appreciate the substantial technical and logistical support by Joe Soluri (Metro Vancouver) in operating the site, and scientific contributions and data provided by C. Reynolds (Metro Vancouver) and S. Howie (Delta, BC).

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

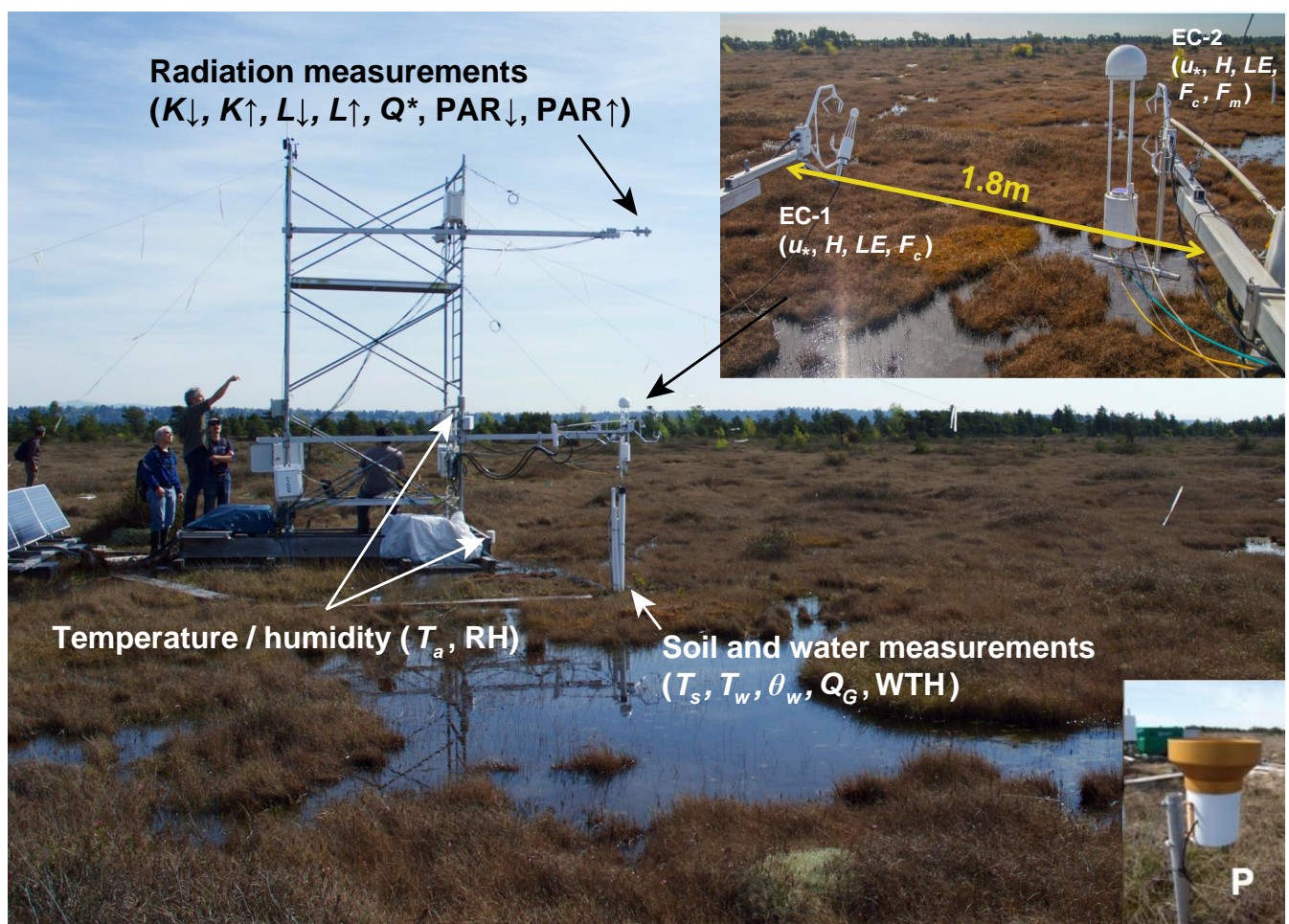

Figure 1: Flux tower on floating platform with EC-1 and EC-2 systems facing south and instruments that measured climate variables indicated (friction velocity ($u_*$), sensible heat flux ($H$), latent heat flux ($LE$), $CO_2$ flux (NEE), $CH_4$ flux ($F_m$), incoming shortwave radiation (K $\downarrow$), outgoing shortwave radiation (K $\uparrow$), incoming longwave radiation (L $\downarrow$), outgoing longwave radiation (L $\uparrow$), net all-wave radiation (Q*), incoming PAR (PAR $\downarrow$), outgoing PAR (PAR $\uparrow$), air temperature ($T_a$), relative humidity (RH), soil temperature ($T_s$), water temperature ($T_w$), soil water content ($\theta_w$), soil heat flux ($Q_G$), water table height (WTH), and precipitation (P)).

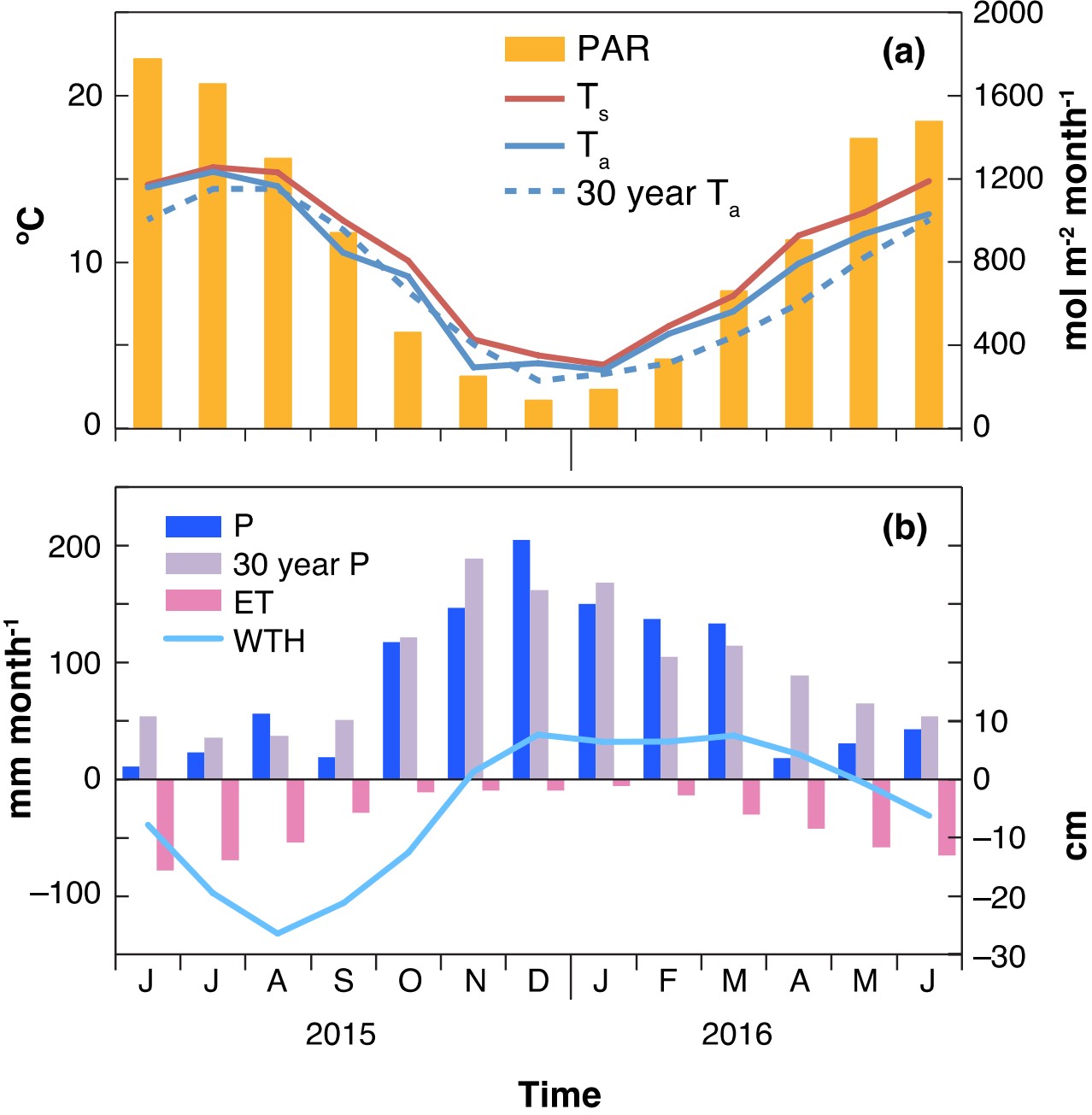

Figure 2: The annual course of weather variables ($T_a$, $T_s$, P, and PAR), ET, and WTH. The 30-year climate normals (30-year $T_a$ and P) were measured at Vancouver International Airport (Data: Environment Canada).

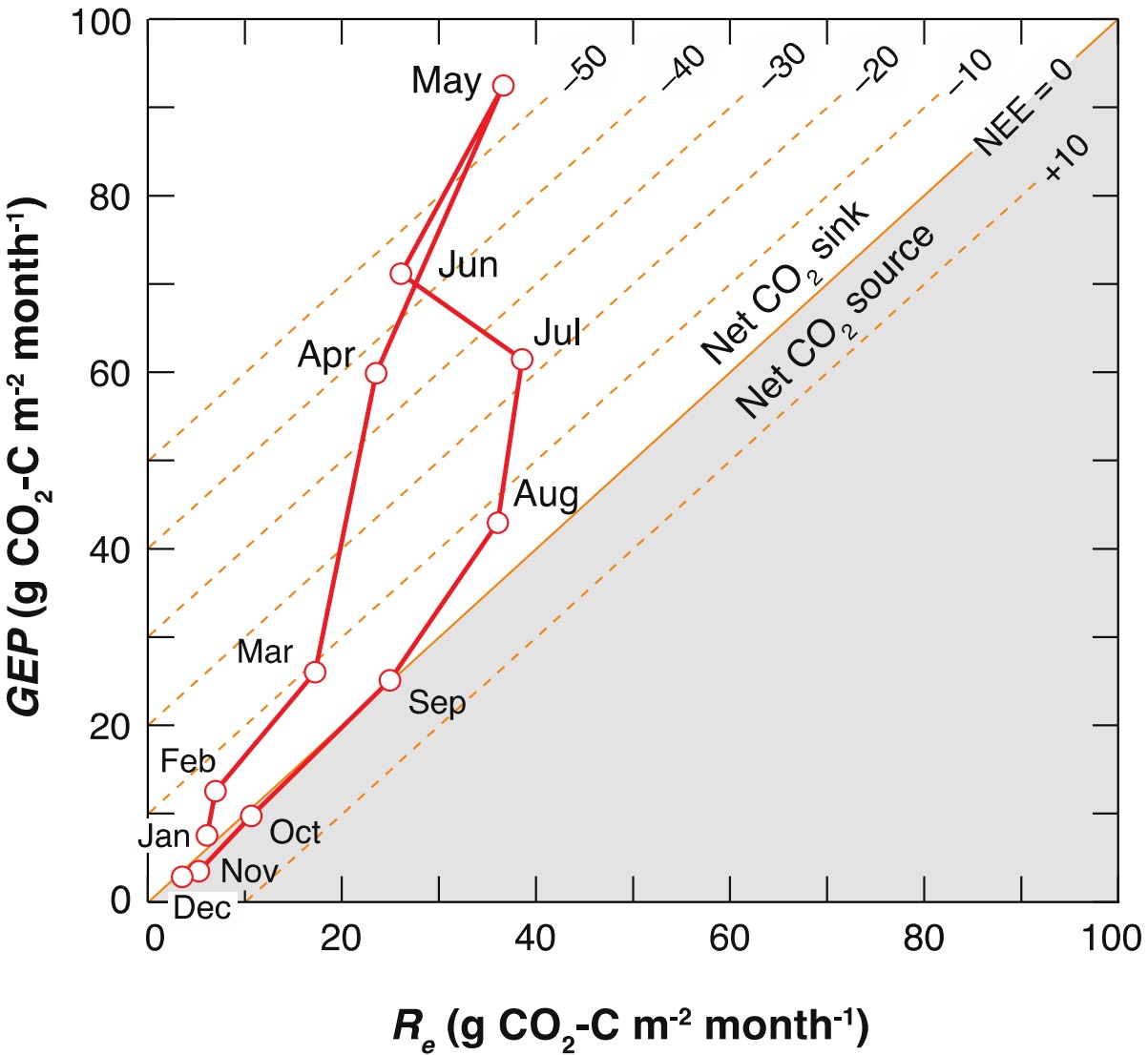

Figure 3: Monthly gap-filled $R_e$ (x-axis) drawn against GEP (y-axis). The resulting NEE can be read off the diagonal lines. The thick 1:1 line shows carbon neutrality, while lines in the upper right are of increasingly negative NEE (uptake) and lines towards the lower right are positive NEE (net source).

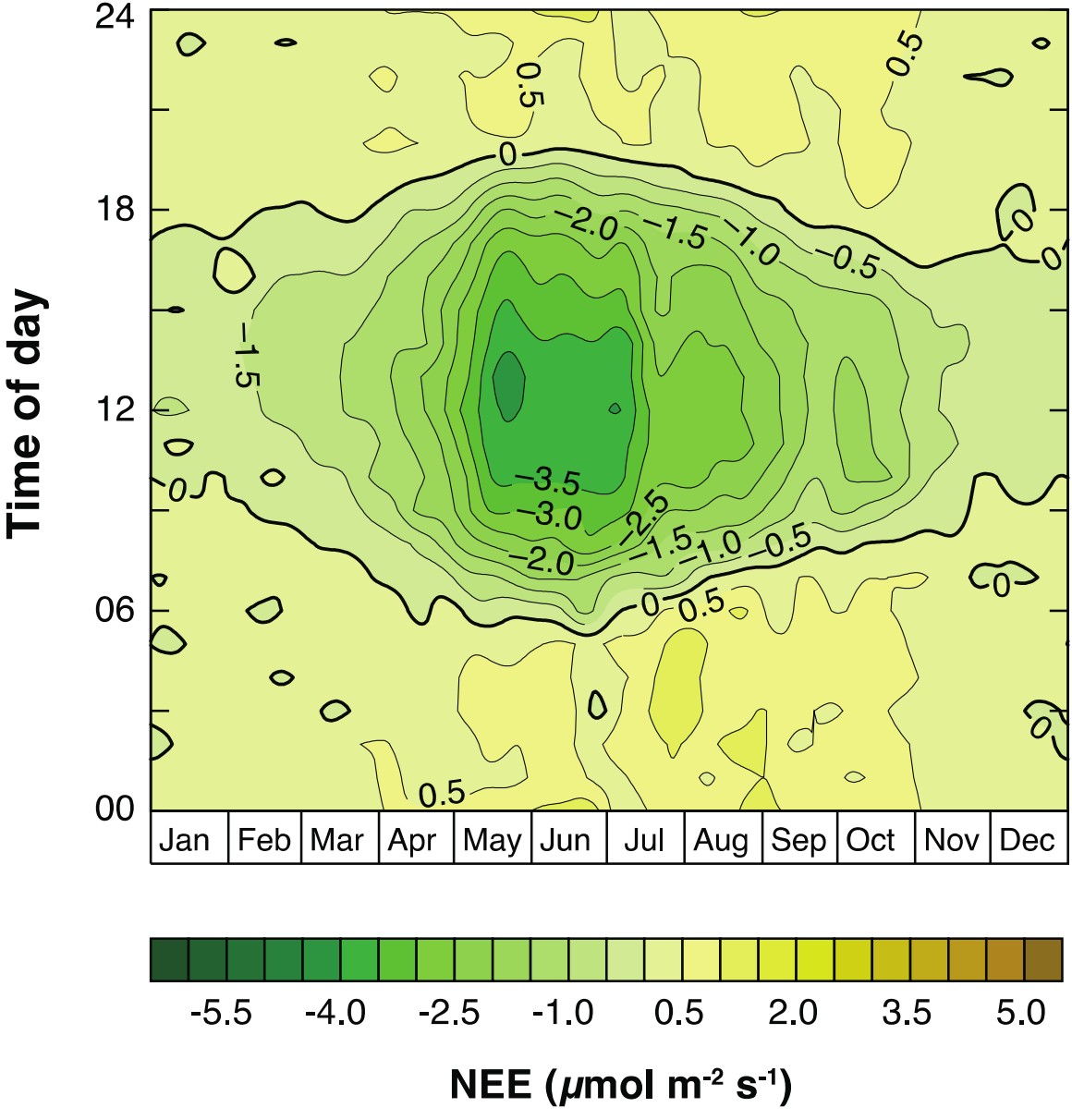

Figure 4: Isopleths of gap-filled NEE (net $CO_2$ fluxes) from the EC-1 system plotted as a composite in the study year. The graph uses a Gaussian filter of $\sigma = 45$ days (which conserves total NEE) to graphically smooth horizontal variations.

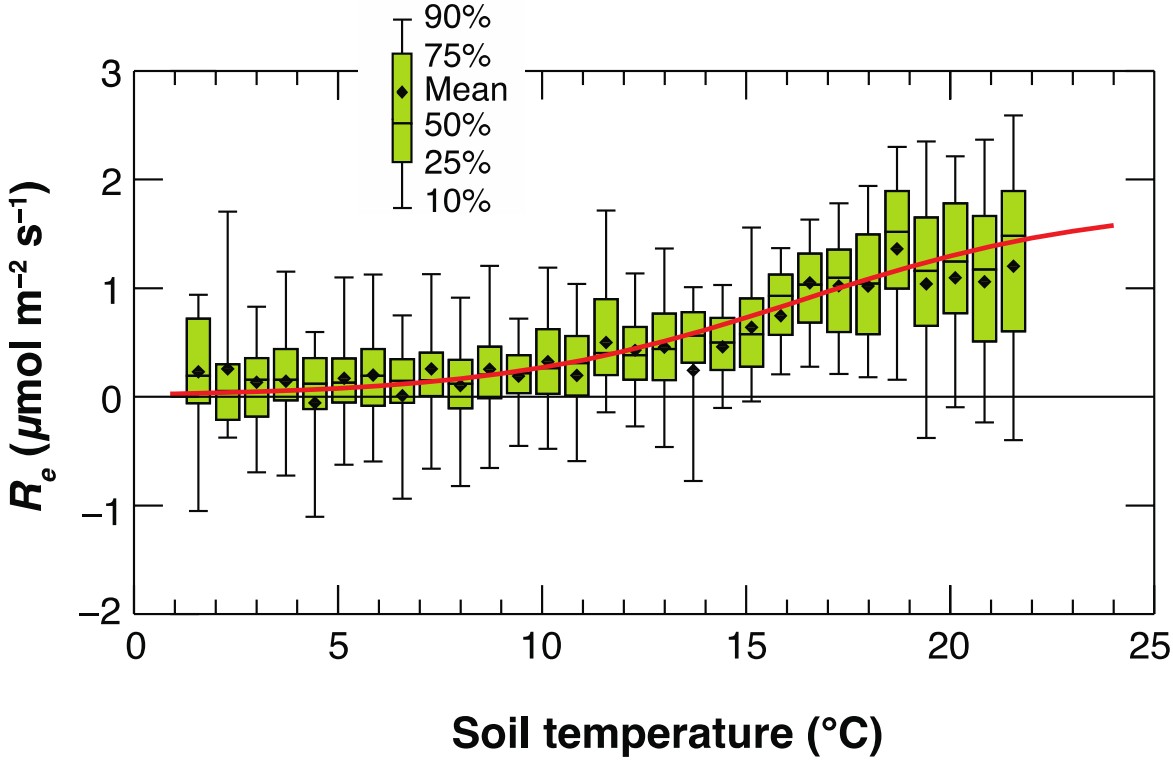

Figure 5: Relationship between $R_e$ (nighttime 30-minute $CO_2$ flux measurements) and $T_{s,5cm}$ during the entire study period. The $u_*$ threshold was 0.08 m s$^{-1}$. The fitted curve is a logistic relationship following Eq. 1. $T_{s,5cm}$ was binned for 32 classes from minimum of $T_{s,5cm}$ to maximum of $T_{s,5cm}$. See Fig. S5 in supplement for seasonal differences. Negative $R_e$ values were caused by measurement uncertainties.

715

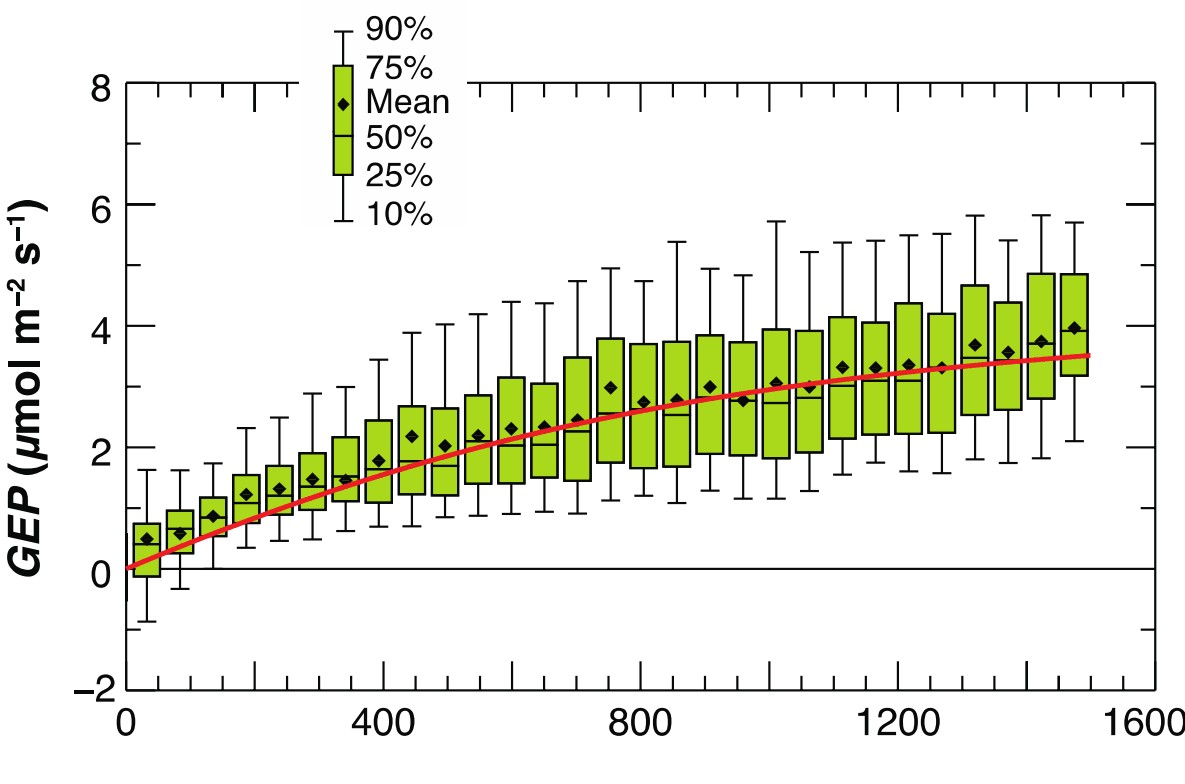

Figure 6: Annual light response curve determined from the daytime 30-minute NEE measurements and Eq. 1, i.e., GEP = $R_e$ + -NEE. The curves are the best fit of the Eq. 2. PPFD was binned for 30 classes from 0 to 1500 μmol m$^{-2}$ s$^{-1}$. Annual $MQY$ was 4.00 mmol C mol$^{-1}$ photons, $P_M$ was 4.68 umol m$^{-2}$ s$^{-1}$, and $C_v$ was 0.7 (fixed).

720

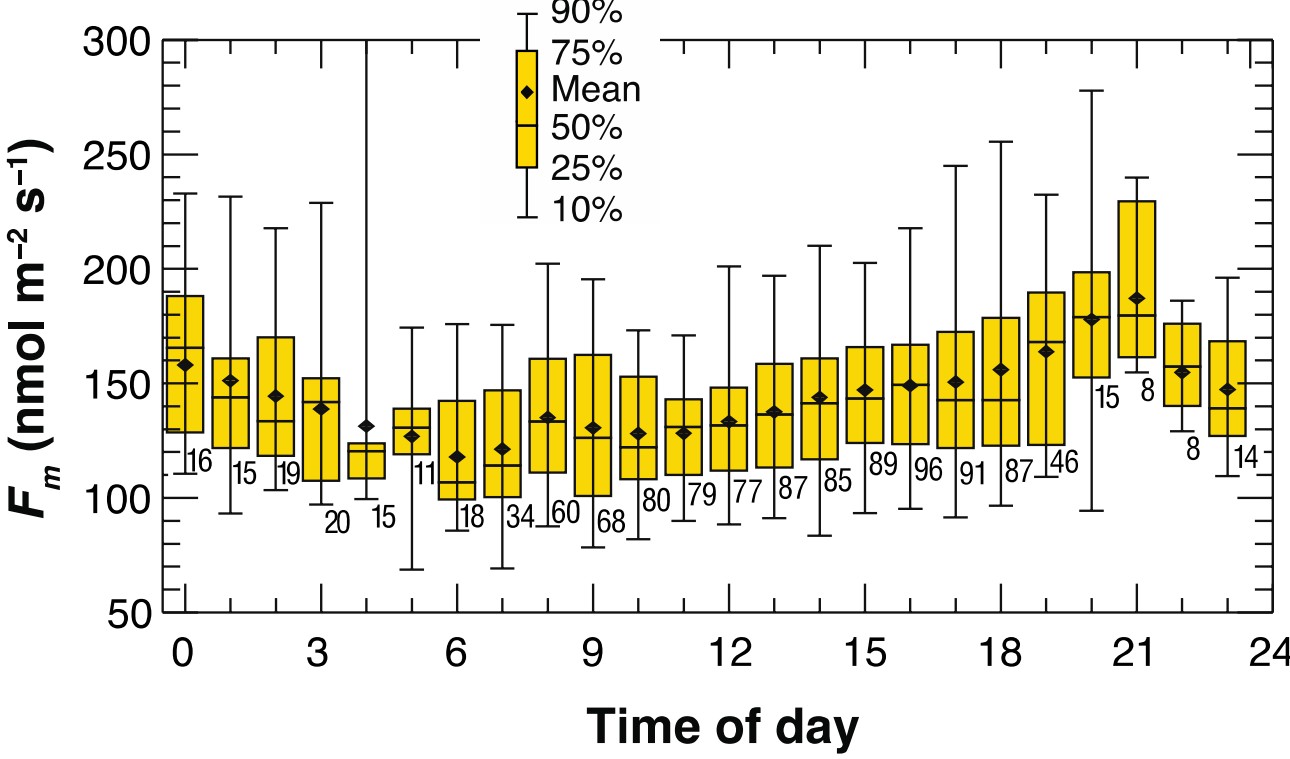

Figure 7: The ensemble-averaged diurnal course of measured $CH_4$ fluxes from the EC-2 system in summer.

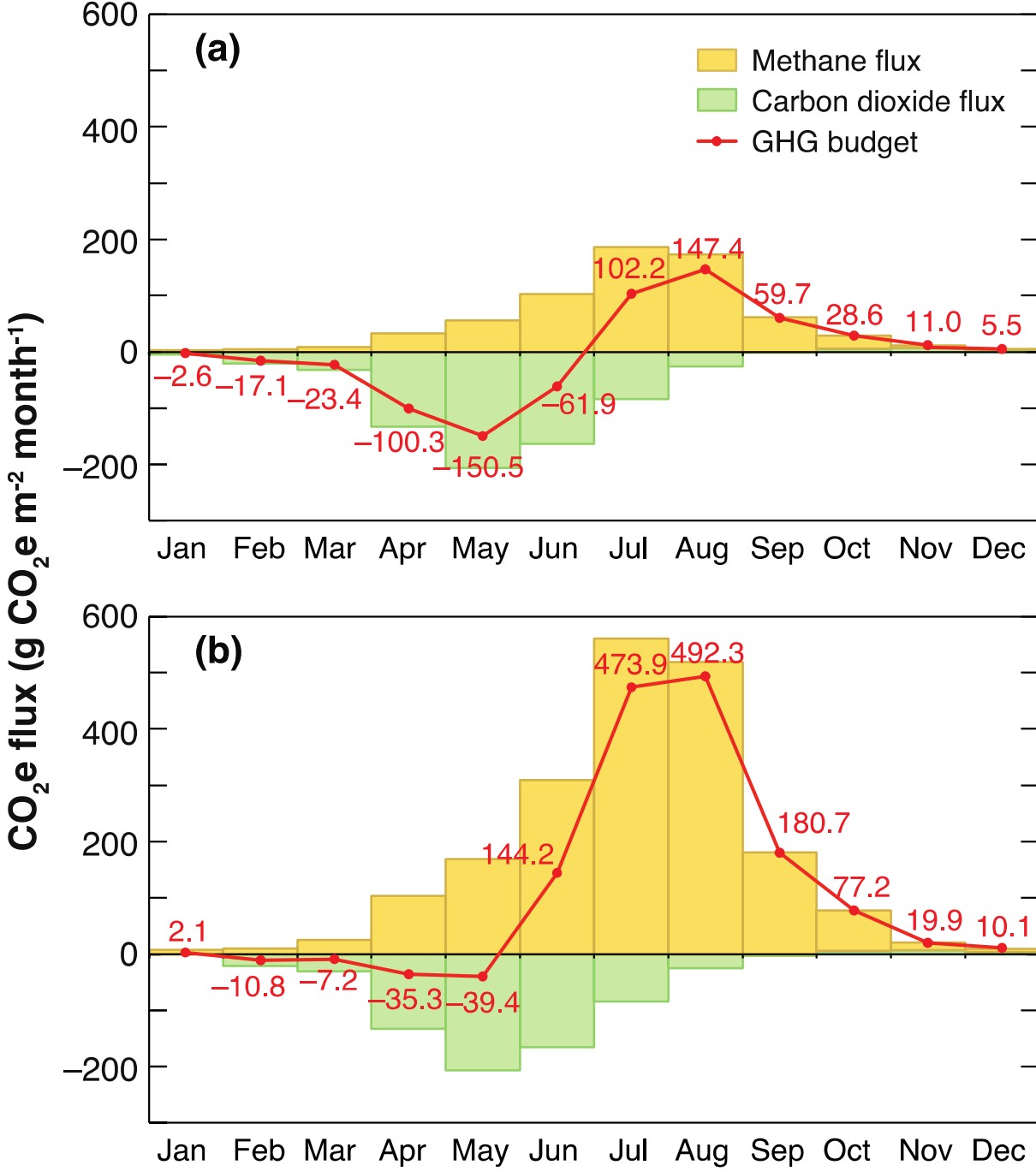

Figure 8: EC-measured monthly $CO_2$, $CH_4$ and net GHGs fluxes shown as $CO_2e$ totals by using (a) 100-year and (b) 20-year GWPs. Missing data were gap-filled.

Table 1: Monthly EC-measured and gap-filled NEE ($CO_2$ fluxes), $CH_4$ fluxes, $CO_2e$ fluxes using 20-year GWP, and $CO_2e$ fluxes using 100-year GWP at the study site during the study period.

| Month | $R_e$ | GEP | NEE | $CH_4$ fluxes | 20-year $CO_2e$ fluxes | 100-year $CO_2e$ fluxes |
|---|---|---|---|---|---|---|
| | (g $CO_2$-C m$^{-2}$ month$^{-1}$) | | | (mg $CH_4$-C m$^{-2}$ month$^{-1}$) | (g $CO_2e$ m$^{-2}$ month$^{-1}$) | (g $CO_2e$ m$^{-2}$ month$^{-1}$) |
| Jan | 6.17 | 7.50 | -1.33 | 93 | 2.06 | -2.57 |
| Feb | 6.94 | 12.46 | -5.52 | 224 | -10.82 | -17.09 |
| Mar | 17.33 | 25.89 | -8.59 | 465 | -7.18 | -23.38 |
| Apr | 23.52 | 59.73 | -36.21 | 1170 | -35.33 | -100.29 |
| May | 36.46 | 92.63 | -56.20 | 1643 | -39.42 | -150.53 |
| Jun | 26.13 | 71.10 | -44.97 | 2670 | 144.23 | -61.85 |
| Jul | 38.53 | 61.47 | -22.94 | 4371 | 474.88 | 102.22 |
| Aug | 36.15 | 42.97 | -6.82 | 3813 | 492.32 | 147.44 |
| Sep | 24.84 | 25.08 | -0.21 | 1650 | 180.67 | 59.71 |
| Oct | 10.76 | 9.58 | 1.18 | 930 | 77.23 | 28.62 |
| Nov | 5.16 | 3.39 | 1.77 | 240 | 19.93 | 10.97 |
| Dec | 3.63 | 2.79 | 0.87 | 155 | 10.13 | 5.50 |
| Study year | g $CO_2$-C m$^{-2}$ year$^{-1}$ | | | g $CH_4$-C m$^{-2}$ year$^{-1}$ | g $CO_2e$ m$^{-2}$ year$^{-1}$ | |
| | 236 ± 16.4 | 415 ± 28.8 | -179 ± 26.2 | 17 ± 1 | 1248 ± 147.6 | -22 ± 103.1 |

Table 2: Comparison of annual NEE, $R_e$ and GEP, over different ecosystems (vegetation covers) in the Vancouver region using EC measurements. Sorted by magnitude of -NEE/GEP ratio.

| Site | Land cover | NEE | $R_e$ | GEP | -NEE/GEP |
|------|-----------|-----|-------|-----|----------|
| | | g C m$^{-2}$ year$^{-1}$ | | | |
| **Burns Bog** **(this study)** Delta, BC | Rewetted raised bog ecosystem | -179 | 236 | 415 | 43% |
| **Westham Island** (CA-Wes)[*] Delta, BC | Unmanaged grassland | -222 | 1215 | 1438 | 15% |
| **Campbell River** (CA-Ca1)[*] Vancouver Island | Douglas-fir forest (~55 yrs) | -328[+] | 1830[+] | 2158[+] | 15% |
| **Buckley Bay** (CA-Ca3)[*] Vancouver Island | Douglas-fir forest (~15 yrs) | 64[+] | 1487[+] | 1423[+] | -4% |

[*] Site identifier in global FLUXNET database (http://fluxnet.ornl.gov).
[+] Data from Krishnan et al., 2009 before fertilisation.