# Peer review of "Annual greenhouse gas budget for a bog ecosystem undergoing restoration by rewetting"

_Biogeosciences, 2016_

## Referee Comment (RC1) · Anonymous Referee #1 · 27 Nov 2016

The authors present a rare and interesting one-year dataset of CH4 and CO2 fluxes for a temperate wetland undergoing restoration by rewetting. Wetlands are hot spots with hot moments and can play over sized roles on the regional greenhouse gas budget, so here, too, more data are welcomed. The site is located on Canada's Pacific Coast. Measurements were made with the aid of eddy-covariance method which allow to estimate surface-atmosphere gas exchange for the whole ecosystem scale. As the measurements of GHG fluxes for bog under restoration are unique the manuscript is worth of publication in Biogesciences. The analysis is thorough, using up-to-date methods. The discussion is comprehensive, showing that the results fit well with those reported from other wetlands.

I suggest authors to consider some improvements, mainly:

[Figure]

1) Estimation of the results uncertainties. The authors estimate the sensitivity of the results on windows size (for Re and GEP). It would nice to estimate the range of results for different gap filling strategies (e.g. neural network) and finally express the annual budget of $CO_2$ in the form NEE= -179±??? g $CO_2$-C m-2 year-1 and similarly for $CH_4$ flux (or at least discuss on the base of recent publications which consider such impact).

2) The gap filling of $CH_4$ is based on regression of the flux against soil temperature. I suggest, to consider to fit parameters of Eq. 3 in the window similar to Re and GEP, not for whole year. The different environmental condition (water table level, vegetation development, temperature of deeper soil levels ect.) can result in different respond of $CH_4$ flux for temperature. The estimation of the parameters in the window would allow to include these influences.

3) The global warming potential (GWP) is the most common measure to asses a combined impact of $CH_4$ and $CO_2$ emission on climate. However, it assumes a pulse emission which is not a case for wetlands, thus the applicability of GWP to asses the role of these ecosystems in the Earth's global radiation budget can be questioned (e.g. Neubauer and Megonigal, 2015; Petrescu et al., 2015). The author could refer to this problem in discussion.

Specific comments

1) L 40 and in other places in text: "wetlands . . . sequester from -146 to -266 g $CO_2$-C m-2 year-1" - negative sequestration means emission? It is easy to guess in this case, especially for those who are familiar with EC measurements, but in general it is not obvious, so one must be careful about a sings of the fluxes (for example nest in the text sequestration in GEP is positive). Please look through the text to clarify.

2) L 265: "In June and July, the fitted curve stayed at 1 $\mu$mol m-2 s-1 because $T_{s,5cm}$ remained above 15oC" – argumentation is not clear for me.

3) L 271: "Two other controls on Re explored were air temperature ($T_a$) and WTH."

Whereas role of WTH is already pointed above (L 268): "Another factor could be the WTH"

4) LL 324-326: Last two sentences in the paragraph seem to be loosely related to the previous.

References:

Neubauer, S., Megonigal, J.P., 2015. Moving beyond global warming potentials to quantify the climatic role of ecosystems. Ecosystems 18, 1000–1013.

Petrescu et al. (2015) The uncertain climate footprint of wetlands under human pressure. Proc. Natl. Acad. Sci. USA, 112, 4594-4599.

---

## Referee Comment (RC2) · Anonymous Referee #2 · 17 Dec 2016

It is interesting to read of research into carbon exchange and the greenhouse gas budget of a part of Burns Bog in coastal BC. Wetland restoration proposals often cite the benefits of reversing the loss of carbon from degrading peat and re-creating a C sink. However, at least in the short-term, emissions of methane from rewetted wetlands can create a net GHG source.

I think that this manuscript is worthy of publication once some of my listed concerns are addressed. My greatest concern is that there is insufficient testing of the results via thorough reference to the wetland flux literature. Specifically, the CO2 flux component (NEE, GEP, Re) magnitudes are compared in detail with results from other types of ecosystems in the region that the authors are familiar with (Table 2), being forests and grassland, but not with relevant wetland studies. While annual NEE indicates a relatively strong CO2 sink, the magnitudes of GEP and Re are rather small. This is certainly

evident from the comparisons on Table 2, and the authors point this out. However, they make no effort to thoroughly test these results against the wetland flux literature. Most reported peatland flux studies are for higher latitude/shorter growing season sites than the present study, so it can be difficult to find relevant comparisons. For instance, the compilation of annual $CO_2$ flux data for "inland wetlands" in Lu et al. (2016) have no GEP and Re data for sites with similar mean annual temperature to Burns Bog. In contrast, there are several sites listed that have GEP and/or Re in the range of magnitudes reported for Burns Bog. Equivalently low GEP were reported for sites with much lower MAT, and therefore short growing seasons, whereas Burns Bog had year-round positive GEP. Similarly low Re have only been reported for sites with very low MAT and extremely low productivity. At Glencar blanket bog in Ireland, with MAT similar to BB but with a short growing season, Sottocornola and Kiely (2010) reported annual GEP and Re of similar magnitude to BB. In summary, to gain some confidence that flux partitioning and calculations have been correctly performed, it is essential that these low GEP and Re at Burns Bog are carefully considered and explained. Similarly, the magnitudes and seasonal patterns of light response parameters and respiration should be compared to the established literature. An appropriate Canadian reference would be Humphreys et al. (2006). The same issue arises for the methane flux. There is a growing body of literature reporting annual and sub-annual FCH4 data from EC sites over wetlands, yet little reference to this literature is made. The authors may have made calculation errors in converting 30-minute fluxes through to annual values, certainly this appears to be the case for the methane fluxes shown in Fig. 6, and listed in Table 1.

Detailed comments

Lines 38-40. Many of the cited studies here are horribly out of date or completely inappropriate. For instance, den Hartog et al. (1994) appears to be only an energy balance study and Schulze et al. (1999) is a forest study. Citing incorrectly at this early stage of a manuscript is a sure way for a reviewer to lose confidence!

Line 40-41. Again, there seems little rationale for choosing these particular references

as representative. Overall, I suggest that the introduction should contain as up-to-date references as possible, especially in the wetland eddy flux discipline where so many recent advances have been made.

Line 46. Details of Mundava reference appears to be incorrect.

Lines 58-59. Poorly written text.

Lines 70-72. The three references supporting this statement about this "other study" appear to be a review followed by two papers describing studies from two different wetlands.

Lines 80-84. No mention of the role of DOC flux contributing to the overall net C flux. Exports of C via DOC can make up a major component. This should be acknowledged in the paper, and a justification made for why it was not assessed.

Section 2, Study area. It would be nice to have some more brief details of BB, such as area, mean annual climate statistics (see later comment).

Line 93. "... highest emissions under a high water table"? Maybe "... associated with high water tables".

Line 105. "... reduced ET as a consequence of senescence." Are there data on this? Reference to another study? Implies a definitive finding, which would be a worthwhile result on its own, but no EC water vapour flux data were presented in the manuscript.

Line 128. The detail that the CSAT3 samples at 60 Hz is unnecessary.

Lines 130-131. Please describe at least whether fluxes were calculated on-line by the dataloggers or during post-processing. It would be useful if the URL for the Crawford et al. report were provided in the reference list.

Line 143. There is no Lee et al. (2016) reference provided, but there is a Lee (2016) MSc thesis. Generally, referring to a thesis should be avoided.

Line 152. Isn't GEP normally defined as gross ecosystem production (i.e. equivalent to GPP)?

Line 165. Range of annual Re: Table 1 lists an even larger value.

Section 3.3.2. Gap filling FCH4. Methane fluxes in wetlands are often the result of a complex interplay of drivers, involving multiple transport pathways and balance between production and oxidation. Moreover, the controls on FCH4 can easily change seasonally and from year to year (Goodrich et al., 2015). I doubt that such a simplistic gap filling procedure as described here is sufficient. This is the reason that multiple-parameter (e.g. Brown et al., 2014) and neural network (e.g. Goodrich et al., 2015) methods are more standard. Therefore, some more convincing details of FCH4 gap filling are required.

Line 190, Eq. 4. Please define the m values for completeness.

Section 4.1. Some comparison of seasonal and annual temperature and precipitation to long-term normals would be useful to justify how close to average (or not) the conditions during the study period were. Also (line 200), I don't believe one can justify listing annual precipitation totals to the precision of one decimal place, given the problems inherent in rain gauges!

Line 210. Why list the author names (Kormann and Meixner) twice?

Line 217. What grasses? Were these wetland species?

General comment: a figure showing the annual course of weather variables and water table would be very useful.

Lines 238-239. The "highest increasing rate of NEE" appears to be from March to April, not May.

Line 242 onwards. It seems of very limited usefulness to compare the wetland fluxes to those from forests and grasslands, and it highlights the completely insufficient comparison with other wetland studies, both for restored peatlands and pristine or disturbed peatlands (see main comment above).

Section 4.3.2. As it stands, Fig. 3 adds nothing to the paper other than a pretty picture. It would be of some use if there was a proper comparison made between these diurnal/seasonal patterns with the literature from other wetlands. FCO2 is only ever used in Fig. 3 and is not properly defined.

Section 4.3.3. Again, the magnitude of Re has not been adequately compared to other wetland flux literature, either on an instantaneous basis or seasonal/annual.

Line 277. I could not find where the measurement of theta_w (moisture content?) was described. Section 4.3.4. Again, this section on GEP is deficient in comparing their values for GEP and various timescales (and light response) with the relevant literature.

Lines 289-290. "We found out there was the light-independent photosynthesis ...". This sentence is rather perplexing. How was this deduced? Also, the PAR range 300-500 is exactly in the range where GEP seems maximally dependent on light (Figs 5, S4)!

Section 4.4.2. Same comment as above about inadequate reference to relevant literature about CH4 fluxes. Lines 296-297. What do "weak" and "significant" mean in the context of CH4 fluxes when the literature is not referred to?

Line 305. Why was it surprising that there was not much of a diurnal course observed for FCH4? The authors seem to be completely unaware of why or why not this flux may or may not follow a diurnal course. Figure 6, with the whole annual period included, would almost certainly mask seasonal differences in diurnal patterns. Also, the units for FCH4 in Fig. 6 is surely incorrect. This should presumably be nmol m-2s-1.

Lines 305-306. "Thermal effects such as recently reported by ...". This is a bit too cryptic. Were the modelling methods of the Poindexter et al. (2016) followed, or is this just an attempt to justify the apparent lack of a diurnal pattern? Besides, at BB the water table was sometimes above the surface and sometimes below, and the annual

vegetation growth changed (as described), so it is logical to assume that a variety of methane transport processes would have operated.

Line 322. By CH4 emissions and CO2 uptake, I presume the CO2-eq values of these are being referred to.

Lines 328-330. This is by no means an adequate way to address the lack of comparison of the CO2 fluxes from this study with the peatland (or other wetland) literature.

Line 371. For peak's sake? Peat?

Figure 6. Units for FCH4 are surely incorrect. If these are actually nmol m-2s-1, a mean flux of around 100 nmol m-2s-1 should yield an annual flux of around 38 g CH4-C m-2yr-1, not the 16 g CH4-C m-2yr-1 as provided in Table 1. The authors should carefully check their flux conversion calculations, for both CH4 and CO2 fluxes, to provide some confidence it has been done correctly.

Figure S1. North orientation should be indicated. Also, note that not all panels show max. contour of 90%.

Figure S3. "Re curves" is not an adequate description. What does it mean "on first day of every two months"? This is not correct.

Figure S4. Same comment about inadequate caption.

References (suggested by this reviewer).

Brown et al. (2014), Evidence for a nonmonotonic relationship between ecosystem-scale peatland methane emissions and water table depth, J. Geophys. Res. Biogeosci., 119, 826–835, doi:10.1002/2013JG002576.

Goodrich et al. (2015), Overriding control of methane flux temporal variability by water table dynamics in a Southern Hemisphere, raised bog, J. Geophys. Res. Biogeosci., 120, doi:10.1002/2014JG002844.

[Figure]

Humphreys et al.     (2006) J. Geophysical Research, 111, G04011, doi:10.1029/2005JG000111.

Lu et al. (2016), Global Change Biology, doi:10.1111/gcb.13424.

Sottocornola and Kiely (2010), Agricultural and Forest Meteorology 150 (2010) 287–297.

---

## Author Comment (AC1) · 7 Jan 2017

**REFEREE COMMENTS:**
**Referee #1**

**MAIN COMMENTS TO THE AUTHOR(S)**
**1) Estimation of the results uncertainties. The authors estimate the sensitivity of the results on windows size (for Re and GEP). It would nice to estimate the range of results for different gap filling strategies (e.g. neural network) and finally express the annual budget of CO2 in the form NEE= -179±??? g CO2-C m-2 year-1 and similarly for CH4 flux (or at least discuss on the base of recent publications which consider such impact).**
[Response]
We appreciate the comments of the referee. The major uncertainties in the annual estimates of GEP, $R_e$, NEE, and $CH_4$ fluxes arise from gap-filling. Therefore, the random uncertainties for GEP, $R_e$, NEE, and $CH_4$ fluxes were calculated using different window sizes for gap-filling. The fixed moving-window method was used. For example, the fitted curve was determined by the data between 60 days into past and 60 days into future when the window size is 120 days. Window sizes of 30, 45, 60, 75, 90, 120, 150, 180, and 365 days were selected for GEP, $R_e$, and NEE. The same selections of window sizes with three additions (210, 240, and 270 days) were applied for estimating the uncertainties in the $CH_4$ budget. However, when the window size was too small, a fitted curve could not be obtained for some periods (e.g. not enough variability in controlling variables or occurrence of data gaps due to weather conditions and power limitations). Any gaps caused by using window sizes too small for modelling GEP, $R_e$, and $CH_4$ fluxes were filled by values obtained using the smallest window sizes that successfully produced a fitted curve. The smallest window sizes that successfully produced valid fitted curves for GEP, $R_e$, and the $CH_4$ budget were 85, 30, and 195 days, respectively.

The average vale and uncertainty of annual GEP, $R_e$, and NEE using all combinations of window sizes were $413 \pm 16$, $234 \pm 10$, and $179 \pm 19$ g C m$^{-2}$ year$^{-1}$, respectively. The annual values of GEP, $R_e$, and NEE from the combinations (90 days for GEP and 120 days for $R_e$) chosen in the manuscript are close to the averages from all combinations. The average value and uncertainty from all different window sizes for annual $CH_4$ budget is $17 \pm 1$ g C m$^{-2}$ year$^{-1}$. Therefore, we decided to use a window size of 365 days for $CH_4$ fluxes to cover the full range of soil temperatures in a single function.

We did not consider additional methods (e.g. neural network approaches) for gap-filling due to limitations in resources. We argue that the method of estimating uncertainties in annual flux measurements by using different gap-filling window sizes should suffice and gives a good idea of the seasonally changing responses to the controls.

**2) The gap filling of CH4 is based on regression of the flux against soil temperature. I suggest, to consider to fit parameters of Eq. 3 in the window similar to Re and GEP, not for whole year. The different environmental condition (water table level, vegetation development, temperature of deeper soil levels ect.) can result in different respond of CH4 flux for temperature. The estimation of the parameters in the window would allow to include these influences.**
[Response]
As mentioned in the response to comment 1, a time-dependent calculation of the response curve was additionally added, and the results are presented in the revised manuscript.

**3) The global warming potential (GWP) is the most common measure to asses a com- bined impact of CH4 and CO2 emission on climate. However, it assumes a pulse emission which is not a case for wetlands, thus the applicability of GWP to asses the role of these ecosystems in the Earth's global radiation budget can be questioned (e.g. Neubauer and Megonigal, 2015; Petrescu et al., 2015). The author could refer to this problem in discussion.**
[Response]
Thank you very much for this valuable suggestion. We agree and add the following statement at the end of Sec. 4.5 (L 324):
"Using GWP to classify a study area as a net GHG source or sink is useful; however, the appropriateness of this method in computing the actual radiative forcing has been questioned (e.g. sustained step-change in $CO_2$ and $CH_4$ fluxes can not be evaluated) and alternative models were proposed (Frolking et al., 2007; Fuglestvedt et al., 2000; Neubauer and Megonigal, 2015; Petrescu et al., 2015; Smith & Wigley, 2000)."

**SPECIFIC COMMENTS TO THE AUTHOR(S)**

**1) L 40 and in other places in text: "wetlands . . . sequester from -146 to -266 g CO2-C m-2 year-1" - negative sequestration means emission? It is easy to guess in this case, especially for those who are familiar with EC measurements, but in general it is not obvious, so one must be careful about a sings of the fluxes (for example nest in the text sequestration in GEP is positive). Please look through the text to clarify.**
[Response]
Thank you very much for the suggestion. First, we removed the minus signs on L 40 as follows:
"Other wetlands around the world sequester from 146 to 266 g $CO_2$-C $m^{-2}$ $year^{-1}$ (Lafleur et al., 2001; Pihlatie et al., 2010; Shurpali et al., 1995)."

Second, we clarified the sign convention and added the following explanation at the end of Sec. 3.3 (L 150):
"In this study, net fluxes of $CO_2$ and $CH_4$ toward the ecosystem surface are negative and net fluxes from the ecosystem surface to the stmosphere are positive. Therefore, negative NEE and $F_m$ represent net $CO_2$ and $CH_4$ uptake, respectively."

**2) L 265: "In June and July, the fitted curve stayed at 1 µmol m-2 s-1 because Ts,5cm remained above 15oC" – argumentation is not clear for me.**
[Response]
Thank you very much for the suggestion. We rephrased the argumentation:
"In June and July, due to general warm condition (>15°C), $R_e$ remained nearly constant at ~1 µmol $m^{-2}$ $s^{-1}$ (the fitted curve stayed in the plateau phase)."

**3) L 271: "Two other controls on Re explored were air temperature (Ta) and WTH." Whereas role of WTH is already pointed above (L 268): "Another factor could be the WTH"**
[Response]
Thank you very much for the suggestion. We modified the sentence as:
"Two other controls on $R_e$ explored were air temperature ($T_a$) and WTH. The role of WTH was described above and $T_a$ had have a similar impact on $R_e$ as $T_{s,5cm}$ when ..."

**4) L 324-326: Last two sentences in the paragraph seem to be loosely related to the previous.**
[Response]
Thank you very much for the suggestion. We decided to delete the last two sentences from L 324 to L 326 for clarity.

---

## Author Comment (AC2) · 22 Jan 2017

**REFEREE COMMENTS:**
**Referee #2**

We greatly appreciate all comments from the reviewer. These detailed comments have greatly improved the quality of the manuscript.

**MAIN COMMENTS TO THE AUTHOR(S)**
**1) My greatest concern is that there is insufficient testing of the results via thorough reference to the wetland flux literature. Specifically, the CO2 flux component (NEE, GEP, Re) magnitudes are compared in detail with results from other types of ecosystems in the region that the authors are familiar with (Table 2), being forests and grassland, but not with relevant wetland studies.**
[Response]
We agree with the comments from the referee and have added the following text on additional comparisons to wetland studies at the end of Section 4.3.1 (at line 254):

"The annual NEE in this study was more negative than in the majority of previously reported NEE values for pristine temperate peatlands, which were weak sinks, typically in the range of -50 g C $m^{-2}$ $year^{-1}$ (Roulet et al., 2007; Christensen et al., 2012; Humphreys et al., 2014; McVeigh et al., 2014; Peichl et al., 2014, Pelletier et al., 2015). Values that are comparable to the current restored wetland were reported in five pristine temperate wetlands: $-248$ g C $m^{-2}$ $year^{-1}$ (Lafleur et al., 2001), $-234$ g C $m^{-2}$ $year^{-1}$ (Campbell et al., 2014), $-210$ g C $m^{-2}$ $year^{-1}$ (Fortuniak et al., 2017), $-189$ g C $m^{-2}$ $year^{-1}$ (Flanagan and Syed, 2011), and $-103$ g C $m^{-2}$ $year^{-1}$ (Lund et al., 2010). The few datasets in the literature for NEE of restored wetlands showed a wide range of values. Some were $CO_2$ sources, with NEE ranging from +103 g C $m^{-2}$ $year^{-1}$ to +142 g C $m^{-2}$ $year^{-1}$ (Strack and Zuback, 2013; Richards and Craft, 2015; Järveoja et al., 2016). Other measurements in restored wetlands, however, were sinks, all of them stronger than in this study, with NEE values ranging from -804 g C $m^{-2}$ $year^{-1}$ to -270 g C $m^{-2}$ $year^{-1}$ (Hendriks et al., 2007; Badiou et al., 2011; Herbst et al., 2013; Knox et al., 2015; Anderson et al., 2016). In this study, values of $R_e$ and GEP were lower than those found for a restored wetland at a comparable latitude in the central Netherlands with slightly lower annual temperature and precipitation (Hendriks et al., 2007). $R_e$ and GEP in this study area were also lower than values for most pristine peatlands at comparable latitudes (Helfter et al., 2015; Levy and Gray, 2015). Comparably low $R_e$ and GEP were reported from the 'Mer Bleue' boreal raised bog (Lafleur et al., 2001; Moore et al., 2002) and from an Atlantic blanket bog (Sottocornola and Kiely, 2010; McVeigh et al., 2014), both of which experienced a lower mean annual temperature."

**2) There is a growing body of literature reporting annual and sub-annual FCH4 data from EC sites over wetlands, yet little reference to this literature is made.**
[Response]
We appreciate reviewer's comments. In order to provide a more comprehensive comparison of our $CH_4$ fluxes, we added the following paragraph to Section 4.4.1 (it starts at line 298):

"The annual $CH_4$ flux in this study area was lower than $CH_4$ fluxes reported for other restored wetlands (Anderson et al., 2016; Hendriks et al., 2007; Knox et al., 2015; Nahlik & Mitsch, 2010). Despite the study area being flooded for most of the study year, $CH_4$ emissions were closer to fluxes measured over drained peatlands (Kroon et al., 2010; Schrier-Uijl et al., 2010). Only Herbst et al. (2013) reported an annual $CH_4$ flux from a restored wetland in Denmark that was lower than in this study (9 to 13 g $CH_4$-C m$^{-2}$ year$^{-1}$). Our annual $CH_4$ flux at 16 g $CH_4$-C m$^{-2}$ year$^{-1}$ was comparable to an average natural temperate wetland $CH_4$ flux, which is typically around 15 g $CH_4$-C m$^{-2}$ year$^{-1}$ (Nicolini et al., 2013; Turetsky et al., 2014; Abdalla et al., 2016; Fortuniak et al., 2017). The $CH_4$ fluxes from a number of temperate and tropical pristine wetlands exceeded the $CH_4$ fluxes reported in this study, including emissions from marshes in the Southwestern US (130 g $CH_4$-C m$^{-2}$ year$^{-1}$, Whiting & Chanton, 2001), tropical wetlands in Costa Rica (82 g $CH_4$-C m$^{-2}$ year$^{-1}$, Nahlik & Mitsch, 2010), and marshes in the Midwestern US (50 g $CH_4$-C m$^{-2}$ year$^{-1}$, Koh et al., 2009). However, all these studies were conducted using chambers and the sampling frequency was at most once per month."

**3) The authors may have made calculation errors in converting 30-minute fluxes through to annual values, certainly this appears to be the case for the methane fluxes shown in Fig. 6, and listed in Table 1.**

[Response]

Thank you very much for bringing this to our attention. In Figure 6, we actually plotted only data that was measured (hence the different number of cases in each hour), and we excluded gap-filled data. There were significantly more datasets available from the summer half-year (higher $CH_4$ fluxes) than from the winter half-year (lower $CH_4$ fluxes), consequently the data in the figure cannot be simply averaged. To make this clearer, we have changed the caption and corrected the units (it was incorrectly labelled "μmol" instead of "nmol"). The corrected Fig. 6 is as follows:

[Figure]

The new caption reads:

"Figure 6:  Diurnal course of  measured $CH_4$ fluxes from the EC-2 system during the study period."

Also, we have corrected the related text in Section 4.4.2 as follows:

"The ensemble diurnal courses of the  $CH_4$ fluxes  measured by the EC-2 system are shown in Fig. 6 from 16[th] June 2015 to 15[th] June 2016."

**SPECIFIC COMMENTS TO THE AUTHOR(S)**
**1) Lines 38-40. Many of the cited studies here are horribly out of date or completely inappropriate. For instance, den Hartog et al. (1994) appears to be only an energy balance study and Schulze et al. (1999) is a forest study. Citing incorrectly at this early stage of a manuscript is a sure way for a reviewer to lose confidence!**
[Response]
We appreciate reviewer's comments. The first paragraph of the introduction has been re-written to include more recent studies and omits den Hartog et al. (1994) and Schulze et al. (1999) as follows:

"Wetland ecosystems play a disproportionately large role in the global carbon (C) cycle compared to the surface area they occupy. Wetlands cover only 6% – 7% of the Earth's surface (Lehner and Döll, 2004; Mitsch et al. 2010), but they act as a major sink for the long-term C storage by sequestering carbon dioxide ($CO_2$) from the atmosphere. For example, strong C sinks (896 to 1139 g $CO_2$-C $m^{-2}$ $yr^{-1}$ and 1236 g $CO_2$-C $m^{-2}$ $year^{-1}$) were found in Southeast USA and Eastern France, respectively (Mitsch et al. 2013; Grasset et al., 2016). Other wetlands around the world sequester around 100 g $CO_2$-C $m^{-2}$ $year^{-1}$ (Petrescu et al., 2015; Bortolotti et al., 2016; Lu et al., 2016). C storage in wetlands has been estimated to be up to 450 Gt C or approximately 20% of the total C storage in the terrestrial biosphere (Bridgham et al., 2006; Lal, 2008; Wisniewski and Sampson, 2012). However, wetlands emit significant quantities of methane ($CH_4$), a powerful greenhouse gas (GHG), due to anaerobic microbial decomposition (Aurela et al., 2001; Rinne et al., 2007). $CH_4$ emissions from wetlands are responsible for 30% of all global $CH_4$ emissions (Bergamaschi et al., 2007; Bloom et al., 2010; Ciais et al., 2013). Peatlands are the most widespread of all wetland types in the world, representing 50 to 70% of global wetlands (Roulet, 2000; Yu et al., 2010). Their dynamics have played an important role in the global C cycle during the Holocene period (Gorham, 1991; Yu, 2011; Menviel and Joos, 2012), and it has been shown that it is crucial to include peatlands in the modelling and analysis of the global C cycle to mitigate the changes in other C reservoirs is highly relevant (Frolking et al., 2009; Wania et al., 2009; Kleinen et al., 2010)."

**2) Line 40-41. Again, there seems little rationale for choosing these particular references as representative. Overall, I suggest that the introduction should contain as up-to-date references as possible, especially in the wetland eddy flux discipline where so many recent advances have been made.**
[Response]
We appreciate reviewer's comments. The Introduction Section has been expanded by adding up-to-date citations (please see the previous response).

**3) Line 46. Details of Mundava reference appears to be incorrect.**
[Response]
We appreciate reviewer's correction. This reference has been discarded to avoid using a thesis as reference, and replaced by Roulet (2000) and Yu et al. (2010).

**4) Lines 58-59. Poorly written text.**
[Response]
We have now rephrased the text in reference to make it clear:
"Additionally, degraded peat increases the risk of peatland fires, which could consequently cause significant $CO_2$ emissions (Gaveau et al., 2014; Page et al., 2002; van der Werf et al., 2004)."

**5) Lines 70-72. The three references supporting this statement about this "other study" appear to be a review followed by two papers describing studies from two different wetlands.**
[Response]
Thank you very much for the suggestion. We have corrected the text as follows:
"In other studies, re-establishing the conditions…"

**6) Lines 80-84. No mention of the role of DOC flux contributing to the overall net C flux. Exports of C via DOC can make up a major component. This should be acknowledged in the paper, and a justification made for why it was not assessed.**
[Response]
We appreciate reviewer's comments. A mention of DOC and its role in net C flux has been made at the end of Section 4.3.1 (at line 254):
"It is important to estimate dissolved organic carbon (DOC) to determine a more complete ecosystem C budget. DOC lost from restored and pristine peatlands have been found typically to range from 3.4 to 16.1 g C $m^{-2}$ $year^{-1}$ (Hendriks et al., 2007; Roulet et al., 2007; Waddington et al., 2008; Koehler et al., 2011), although, Chu et al. (2014) reported a net DOC import for a marsh of $23 \pm 13$ g C $m^{-2}$ $year^{-1}$. D'Acunha et al. (2016) estimated DOC export for the current study area for Jan – Dec 2016 to be 22.4 g C $m^{-2}$ $year^{-1}$ (15% of annual NEE)."

**7) Section 2, Study area. It would be nice to have some more brief details of BB, such as area, mean annual climate statistics (see later comment).**
[Response]
Thank you very much for the suggestion. We added the information at line 86 and line 101 as follows:
"Burns Bog in Delta, BC, on Canada's Pacific Coast, is part of a remnant peatland ecosystem that is recognized as the largest raised bog ecosystem (2,042 ha) on North America's west coast."
"… bracing (Howie et al., 2009). Based on the weather data for 1981 to 2010 from the closest Environment Canada weather station, Vancouver International Airport, the average annual temperature was 10.4 ℃ and average annual precipitation was 1189 mm. Following rewetting, …"

**8) "... highest emissions under a high water table"? Maybe "... associated with high water tables".**

[Response]

Thank you very much for the suggestion. The suggested correction has been made:

"… highest emissions associated with high water tables."

**9) "... reduced ET as a consequence of senescence." Are there data on this? Reference to another study? Implies a definitive finding, which would be a worthwhile result on its own, but no EC water vapour flux data were presented in the manuscript.**

[Response]

Yes. We have continuous ET data which were gap-filled using REddyProc (Max Planck Institute for Biogeochemistry). Monthly ET values have been added to the figure showing the annual course of weather variables:

[Figure]

To make it further clear, we have now added more details at line 104 as follows:

"In September and October, a water table rise due to the increase in precipitation and reduced evapotranspiration (ET) as a consequence of reduced available energy and senescence of sedges was observed, which is similar to water table observations in other temperate wetlands (Lafleur et al., 2005; Rydin and Jeglum, 2006)."

**10) The detail that the CSAT3 samples at 60 Hz is unnecessary.**
[Response]
We appreciate reviewer's suggestion. This information was edited as follows (at line 127):
"The CSAT-3 measured the longitudinal, transverse and vertical components of the wind vector and sonic temperature and output data at 10 Hz."

**11) Lines 130-131. Please describe at least whether fluxes were calculated on-line by the dataloggers or during post-processing. It would be useful if the URL for the Crawford et al. report were provided in the reference list.**
[Response]
We appreciate reviewer's suggestion. The fluxes were calculated in the post-processing, and this information has been added as follows:
"… were calculated in post-processing of 30-min data blocks following the procedures documented in Crawford et al. (2013)."
Also, the permanent link (http://hdl.handle.net/2429/45079) was added in the reference list.

**12) Line 143. There is no Lee et al. (2016) reference provided, but there is a Lee (2016) MSc thesis. Generally, referring to a thesis should be avoided.**
[Response]
We appreciate reviewer's suggestion. As suggested, reference to the Lee (2016) thesis has been removed:
"Gaps in the climate data (<1% of the year) were filled using measurements at nearby climate stations."

**13) Line 152. Isn't GEP normally defined as gross ecosystem production (i.e. equivalent to GPP)?**
[Response]
Yes, GEP usually stands for gross ecosystem production or productivity, which is equivalent to gross primary production (GPP). GEP can also stand for gross ecosystem photosynthesis which is equivalent to gross ecosystem productivity. In order to be consistent, we modified the definition in Section 3.3.1 (line 153) as follows:
"…and gross ecosystem productivity (GEP), i.e. NEE = $R_e$ − GEP."
Also, the name of Section 4.3.4 was corrected to:
" 4.3.4 Gross ecosystem productivity"

**14) Line 165. Range of annual Re: Table 1 lists an even larger value.**
[Response]
Thank you very much for pointing out the discrepancy. The sensitivity test of window sizes on gap-filling was re-run on a more comprehensive scale based on comments from Referee #1, as a result of which this sentence has been modified as follows:
"However, the sensitivity of choosing different window sizes on gap-filled $R_e$ was small, varying the annual value between 226 and 245 g C m$^{-2}$ year$^{-1}$."

**15) Section 3.3.2. Gap filling FCH4. Methane fluxes in wetlands are often the result of a complex interplay of drivers, involving multiple transport pathways and balance between production and oxidation. Moreover, the controls on FCH4 can easily change seasonally and from year to year (Goodrich et al., 2015). I doubt that such a simplistic gap filling procedure as described here is sufficient. This is the reason that multipleparameter (e.g. Brown et al., 2014) and neural network (e.g. Goodrich et al., 2015) methods are more standard. Therefore, some more convincing details of FCH4 gap filling are required.**

[Response]

We appreciate reviewer's suggestion. We have tested the effects of all other possible controls including WTH, $\theta_w$, oxidation reduction potential, and $T_a$ on $CH_4$ fluxes. There was no relationship between these variables and $CH_4$ fluxes. We were forced to use the relationship between $T_s$ and $CH_4$ fluxes. The strongest relationship was an exponential one with an $R^2$ value of 0.66 (logarithmic, linear and polynomial relationships resulted in $R^2$ values of 0.46, 0.52 and 0.54, respectively).

**16) Line 190, Eq. 4. Please define the m values for completeness.**

[Response]

We appreciate reviewer's suggestion. The *m* values have been included:

"… , $m_{CO_2}$ is the molecular mass of $CO_2$ (44.01 g mol$^{-1}$), and $m_{CH_4}$ is the molecular mass of $CH_4$ (16.04 g mol$^{-1}$)."

**17) Section 4.1. Some comparison of seasonal and annual temperature and precipitation to long-term normals would be useful to justify how close to average (or not) the conditions during the study period were. Also (line 200), I don't believe one can justify listing annual precipitation totals to the precision of one decimal place, given the problems inherent in rain gauges!**

[Response]

We appreciate reviewer's suggestion. First, we reduced the significant digits of annual precipitation totals to 0. Second, monthly precipitation and temperature measured during the study year at the tower and over 30 years at Vancouver International Airport were plotted in the figure showing the annual course of weather variables:

[Figure]

**18) Line 210. Why list the author names (Kormann and Meixner) twice?**

[Response]

We removed one of them and rewrote as follows:

"… using an analytical turbulent source area (turbulent footprint) model  (Kormann and Meixner, 2001)

**19) Line 217. What grasses? Were these wetland species?**
[Response]
Yes, the common name of the dominant plant species (*Rhynchospora alba*) mentioned in Section 2 is white beak-sedge. The explanation has been added to Section 4.2.2 for clarity:
"Mosses and white beak sedge (the common name of *Rhynchospora alba*) started to grow …"

**20) General comment: a figure showing the annual course of weather variables and water table would be very useful.**
[Response]
We appreciate reviewer's suggestion. A new figure was made (see our response to Comment 9 above).

**21) Lines 238-239. The "highest increasing rate of NEE" appears to be from March to April, not May.**
[Response]
This sentence has been re-written as follows for clarity:
"The highest rate of increase in the magnitude of NEE and the highest magnitude of NEE both occurred early in growing season (Fig. 2)."

**22) Line 242 onwards. It seems of very limited usefulness to compare the wetland fluxes to those from forests and grasslands, and it highlights the completely insufficient comparison with other wetland studies, both for restored peatlands and pristine or disturbed peatlands (see main comment above).**
[Response]
We appreciate reviewer's suggestion. This comparison gives us information on how different the C exchange of a wetland is compared to other ecosystems in the same region, sharing the same climatic conditions. However, we have now added a detailed discussion comparing this study to other pristine and restored wetlands as follows. See our response to Comment 1 above.

**23) Section 4.3.2. As it stands, Fig. 3 adds nothing to the paper other than a pretty picture. It would be of some use if there was a proper comparison made between these diurnal/seasonal patterns with the literature from other wetlands. FCO2 is only ever used in Fig. 3 and is not properly defined.**

[Response]

We appreciate reviewer's comment. The label of scale ($FCO_2$) has been corrected to Fc for clarity. Figure 3 is the only place where detailed diurnal and seasonal trends in $F_C$ are shown, which are valuable data and evidence to support our conclusions. To improve readability, we have now added the following information about Fig 3 (at line 256):

"The seasonally-changing diurnal course of gap-filled NEE with isopleths over time of day and year is shown in Fig. 3. The daily maximum in GEP changed with season resulting in the high magnitude of NEE during midday between May and July (~ -3.5 $\mu$mol m$^{-2}$ s$^{-1}$) with the highest magnitude of NEE occurring in May. Nighttime NEE, i.e., $R_e$, showed relatively small variation with season, and on average was ≤1 $\mu$mol m$^{-2}$ s$^{-1}$ for most of the study period. The rapid decrease in monthly $R_e$ from May to June was caused by low $R_e$ in early morning or at nightfall in June."

**24) Section 4.3.3. Again, the magnitude of Re has not been adequately compared to other wetland flux literature, either on an instantaneous basis or seasonal/annual.**

[Response]

We appreciate reviewer's suggestion. A detailed discussion comparing $R_e$ from this study to other pristine and restored wetlands has been added at line 254 (see our response to Comment 1).

**25) Line 277. I could not find where the measurement of theta_w (moisture content?) was described. Section 4.3.4. Again, this section on GEP is deficient in comparing their values for GEP and various timescales (and light response) with the relevant literature.**

[Response]

We appreciate reviewer's suggestion. The information on the measurement of soil volumetric water content has been added to Section 3.1:

"A soil volumetric water content ($\theta_w$) sensor (CS616, CSI) was inserted vertically to measure integrated $\theta_w$ from the surface to a depth of 0.30 m."

**26) Lines 289-290. "We found out there was the light-independent photosynthesis ...". This sentence is rather perplexing. How was this deduced? Also, the PAR range 300-500 is exactly in the range where GEP seems maximally dependent on light (Figs 5, S4)!**
[Response]
We appreciate reviewer's suggestion. The second paragraph in Section 4.3.4 was re-written for clarity as follows:

"Other possible controls on GEP explored were WTH and $T_a$. We found that WTH was not a control on GEP in the current study as the study area remained fairly wet throughout the year. Furthermore, the effects of $T_a$ on GEP were approximately limited between 10 and 15 ℃."

**27) Section 4.4.2. Same comment as above about inadequate reference to relevant literature about CH4 fluxes. Lines 296-297. What do "weak" and "significant" mean in the context of CH4 fluxes when the literature is not referred to?**
[Response]
We appreciate reviewer's suggestion. We re-wrote the sentence at line 298 for clarity:
"Seasonally, it was a weaker $CH_4$ source in fall than in summer … ."

**28) Line 305. Why was it surprising that there was not much of a diurnal course observed for FCH4? The authors seem to be completely unaware of why or why not this flux may or may not follow a diurnal course. Figure 6, with the whole annual period included, would almost certainly mask seasonal differences in diurnal patterns. Also, the units for FCH4 in Fig. 6 is surely incorrect. This should presumably be nmol m-2s-1.**

[Response]

We appreciate reviewer's suggestion. In Figure 6, we actually plotted only data that were measured, i.e., gap-filled data were excluded (hence the different number of cases in each hour). There were significantly more data available from the summer half-year (higher $CH_4$ fluxes) than from the winter half-year (lower $CH_4$ fluxes). Therefore, we edited the text in Section 4.2.1 starting at line 307 to line 309 as follows:

"Surprisingly, there was only small diurnal variation observed for $CH_4$ fluxes in the summer months, as has been found in other studies (Juutinen et al., 2004; Wang and Han, 2005; Long et al., 2010; Su et al. 2013). In the current study area, with changes in WTH and vegetation growth occurring during the year, there were likely several processes affecting $CH_4$ transport, which masked the diurnal pattern of $CH_4$ fluxes. Furthermore, $T_{s,5cm}$ appeared to be the main environmental control on $CH_4$ fluxes in this study but did not have as strong effect on $CH_4$ emissions as found in previous studies. Thus $CH_4$ was continuously emitted at a similar rate during daytime and nighttime.  From January to March and October to December, the winter half-year, the study site had constant $CH_4$ emissions of less than 50 nmol $m^{-2}$ $s^{-1}$, and almost no diurnal variation was observed. July had the greatest $CH_4$ emissions, and the highest magnitude (>150 nmol $m^{-2}$ $s^{-1}$) appeared in the evening (3 pm to 9 pm). This corresponded to the lagged effect of soil temperature and may be partly due to convective turbulent mixing caused by cooling during the evening (Gordwin et al., 2013)."

Thank you for pointing out the error in the units in Fig. 6. We have corrected the units to nmol.

**29) Lines 305-306. "Thermal effects such as recently reported by ...". This is a bit too cryptic. Were the modelling methods of the Poindexter et al. (2016) followed, or is this just an attempt to justify the apparent lack of a diurnal pattern? Besides, at BB the water table was sometimes above the surface and sometimes below, and the annual vegetation growth changed (as described), so it is logical to assume that a variety of methane transport processes would have operated.**

[Response]

We appreciate reviewer's suggestion. This reference was discarded for clarity, and the discussion of the diurnal course of $CH_4$ fluxes was added, please refer to our response to the previous comment.

**30) Line 322. By CH4 emissions and CO2 uptake, I presume the CO2-eq values of these are being referred to.**
[Response]
Thank you very much for the suggestion. We changed the text as follows:
"In short, the critical time period for both, $CO_2$ and $CH_4$ fluxes in terms of $CO_2e$, was the growing season when magnitude of fluxes changed differently across the growing season. "

**31) Lines 328-330. This is by no means an adequate way to address the lack of comparison of the CO2 fluxes from this study with the peatland (or other wetland) literature.**
[Response]
Thank you very much for the suggestion. A detailed discussion comparing this study to other pristine and restored wetlands has been added at line 254 (see our response to Comment 1).

**32) Line 371. For peak's sake? Peat?**
[Response]
This error has been corrected as follows:
"Chestnutt, C.: For peat's sake: A water …"

**33) Figure 6. Units for FCH4 are surely incorrect. If these are actually nmol m-2s-1, a mean flux of around 100 nmol m-2s-1 should yield an annual flux of around 38 g CH4-C m-2yr-1, not the 16 g CH4-C m-2yr-1 as provided in Table 1. The authors should carefully check their flux conversion calculations, for both CH4 and CO2 fluxes, to provide some confidence it has been done correctly.**
[Response]
Thank you very much for the suggestion. In Figure 6, we plotted only data that were measured (as we indicated in our response to Comment 28). There were significantly more data available from the summer half-year (higher $CH_4$ fluxes) than from the winter half-year (lower $CH_4$ fluxes), consequently the data in the figure cannot be simply averaged. We have changed the caption accordingly and corrected the unit (we incorrectly used "µmol" instead of "nmol"). The new caption reads:
"Figure 6: (a) Diurnal course of filled measured $CH_4$ fluxes from the EC-2 system during the study period."

Also, we have corrected the related text in Section 4.4.2 as follows:
"The ensemble diurnal courses of the gap-filled $CH_4$ fluxes (measured $CH_4$ emissions and gap-filled by modelled $CH_4$ fluxes) measured by the EC-2 system are shown in Fig. 6 from 16[th] June 2015 to 15[th] June 2016."

**34) Figure S1. North orientation should be indicated. Also, note that not all panels show max. contour of 90%.**

[Response]

Thank you very much for the suggestion. The fact that not all panels show the 90% contour line is intentional. All source areas were calculated as gridded data for a 1 x 1 km box (open source code see https://github.com/achristen/Gridded-Turbulent-Source-Area). If a contour line for a certain probability reaches the border of the model domain, the exact shape of the probabilities outside the domain are unknown, and hence the contour cannot be drawn, even within the domain. The new figure was drawn for including north orientation and vegetation conditions in different seasons:

[Figure]

**35) Figure S3. "Re curves" is not an adequate description. What does it mean "on first day of every two months"? This is not correct.**
[Response]
Thank you very much for the suggestion. The new caption for Fig. S3 reads:
"Boxplots of measured $R_e$ (nighttime NEE) plotted against $T_{s,5cm}$ with a fitted curve on the first day of each time period using a window size of 120 days."

**36) Figure S4. Same comment about inadequate caption.**
[Response]
Thank you very much for the suggestion. The new caption for Fig. S4 reads:
"Light response curves on the first day of each time period using a window size of 90 days."

---

## Referee Report (RR1)

**Review of *Annual greenhouse gas budget for a bog ecosystem undergoing restoration by rewetting* by Lee et al. (first revision)**

In general the authors have made appropriate changes to address many of my comments on their first submission. I have two substantial comments followed by a number of more minor ones.

**Substantial**

There has been no attempt to discuss or quantify the likely size of uncertainties of the fluxes, especially as they relate to annual fluxes and $CO_2$-e values. It is not really acceptable to publish these data now without considering the likely size of uncertainties. Given my comments below about gap filling CH4 fluxes, I strongly suggest that a due consideration of the magnitude of uncertainties at the annual timescale is warranted.

My comments about gap filling CH4 fluxes have not really been adequately addressed. I suggest that the nature of the relationship settled on in 3.3.2 be supported by a figure, even if it is in supplementary material. I do find it surprising that an r2 as high as 66% was found, unless it is simply the result of the extremely wide range of flux values. If that were the case, I'd be more concerned about r2 within the region of the relationship where most of the annual flux was derived (i.e. during warm temperatures). This is also why an estimate of uncertainty is required. Moreover, the authors hinted that little CH4 flux data were available in winter, increasing uncertainty. They also hint (lines 328-329) that there may well have been hysteresis in the temperature vs Fm relationship. It may well be more defensible to gap fill using the means of daily 30-minute measurements to construct daily means. But without seeing the data, it's difficult to judge. Suggest looking at Goodrich et al. (2015) JGR-BG (DOI: 10.1002/2014JG002844) for a situation with strong hysteresis in methane fluxes from a wetland, where a "simple" regression-based approach was demonstrated to be inadequate.

**Detailed comments**

Throughout: "eddy-covariance" should be "eddy covariance". The hyphen is unnecessary.

Line 1 of abstract: peatlands have been drained for many other purposes that have also changed them into C sources.

Introduction and lines 38-40. I suggest that the authors might consider constraining their introduction (or at least detailed examples of fluxes) to peatlands. Wetlands are so incredibly diverse that it is almost meaningless to provide context for a study focused on a restored peatland. The Grasset et al. (2016) paper seems a rather strange choice because this appears to have been an aquatic study and not even focussed on annual fluxes. I couldn't find the CO2-C values cited in this or the Mitsch et al. (2013) papers. It is really important at this stage of a manuscript that the authors concentrate on verifying their source material and providing relevant and authoritative literature sources.

Line 73 onward. Using the term "long-term" suggests multi-year. Clearly this paper is not multi-year so, at least in the conclusion, the authors might need to acknowledge the additional uncertainties affecting their conclusions caused by there being a single year of study.

Lines 130-131. The change to the 60 Hz detail has not been made.

Line 146. "Small gaps … and H2O fluxes"?

Line 154 "atmosphere" is mis-spelled.

Use of terms Fc and NEE. In general there is inconsistent use of these terms which, to an unfamiliar reader, might be very confusing. For example, why on line 155 use "NEE and Fm" rather than "Fc and Fm"?

Lines 167-168. It is not really accurate to state that the parameters "were determined separately for each day of the year", when the dataset for each day consisted of a window of 120 days width! Consider a more accurate description.

Lines 173-182. This does not actually describe completely how GEP was modelled. The first step, surely, was to partition GEP out from measured values of NEE (using modelled Re), and then the light response function was fitted.

Line 207. Include reference to (the new) Fig. 2.

Line 246. "Despite…". It is not quite clear what this sentence means. Maybe replace with "Compared to …".

Line 247. This should be a reference to Fig. 3, not Fig. 4.

Line 264. Replace "pristine" with "more or less pristine". For example, the Campbell et al. (2014) study was for a drainage impacted peatland. It is hard to judge what "pristine" actually means, and so most wetland scientists will avoid the term.

Lines 277-281. There needs to be some details provided about the methodology used to calculate DOC fluxes, since the reference is unpublished. At least it is nice to see some uncertainty value provided!

Line 284. Where is the "daily maximum in GEP" shown?

Lines 286-287. "The rapid decrease in monthly Re (Figure 3?) from  May to June …". Fig. 3 does not support this statement.

Lines 319-321. How were these controls investigated? How did you determine "the effects of Ta on GEP were approximately limited between 10 and 15 C."? (The underlined words make no sense!) It is notoriously difficult to isolate the effects of temperature versus light quality versus VPD, because they are all strongly correlated.

Line 326. Still using the inaccurate term "significant". Do you mean "Much larger"?

Lines 328-329. Hysteresis in the Ta vs Fm relationship is hinted at. This suggests that the simplistic gap filling method might be quite inaccurate at these times.

Lines 346-347. Why is this surprising when it has been reported by many other studies?

Line 365. "… magnitude of fluxes changed differently …" is fairly meaningless. What do you mean?

Lines 366-367. Why should they be expected to be representative?

Lines 368-371. This paragraph is a very weak response to Reviewer 1's comment. It is barely coherent. It does not address the long-term nature of C accumulation (from CO2) in wetlands vs the short lifetime of CH4 in the atmosphere.

Line 383. It is surely unusual that there is such a huge seasonal variation in Fm, especially given the temperate climate and high water table in winter. Can the authors suggest why this might be the case? The proper place for this is in Section 4.4.1.

Line 387. If the flux of DOC is taken into account, the annual C sink strength is less than this.

Figure 2. Caption does not mention ET. Was it measured at the site and gapfilled or is it estimated from climate station values?

Figure 3 caption. The x-axis label (F_CO2) has not been corrected.

Figure 7 caption is inadequate. Insert "Ensemble". Note that mainly summer is represented.

Figure S3. It would be preferable to use a common x-axis scale on all of these panels (as was done for Fig S4).

Figures S3 and S4. It is still a little confusing what "on the first day of each time period" means. Presumably this is the function derived for the first day of the period – which is actually derived from data sourced 60 days prior to 60 days after. Surely it would make more sense to choose a day close to the center of the time period.

---

## Referee Report (RR2)

Review of *Annual greenhouse gas budget for a bog ecosystem undergoing restoration by rewetting* by Lee et al. (second revision)

In this review I have focussed only on the appropriateness of changes made in response to my previous two reviews. In general the authors have made appropriate changes to address my previous comments. Most of my new comments listed below are of a more minor nature and I do not wish to review this manuscript again.

David Campbell, 22 April 2017.

**Minor comments**

Line 149. "Longer gaps in CO2 and CH4 fluxes were filled …"

Line 155. Valid data for EC-2 being for just 32% of the year. This needs more careful explanation and consideration. Later on it is implied that most of the missing data were for winter periods. Please provide a very brief summary of missing data by season. How does this affect the annual estimate? See also comment on Fig. 7.

Line 159. $CO_2$, not $C0_2$. Replace "(e.g." with "(i.e."

Line 161 "(GEP)" is unnecessary here. Suggest delete.

Line 168. "r2 decides…" Replace "decides" with "determines".

Line 2016. What approach was taken for gap filling $CH_4$ fluxes (or constraints on this approach) for parts of the year with large amounts of missing fluxes? See comments for Line 155, where the distribution of missing flux data not fully described, but implied that winter fluxes largely absent.

Line 326. Missing word "effect of missing".

Section 4.3.1. Please include uncertainties for annual flux values in this text, or refer to relevant table.

Line 275. "early in the growing season".

Line 296. Cited annual NEE of -804 g C m$^{-2}$. This is a highly unlikely annual value. I do not have the time to check on all these references but suspect that there will be issues with methods or extrapolation to annual values. Could the authors please ensure that they consistently use the literature, e.g. don't mix chamber and EC studies, or at least note where very low confidence exists in some of these values.

Line 303. Please remove quote marks around Mer Bleue. This is a proper noun.

Line 304. Add "… than Burns Bog" or similar at the end of this sentence.

Line 305. 'dissolved organic carbon (DOC) export …"

Line 309. The headspace equilibration technique is not used to estimate DOC concentration. Please read the AGU poster abstract (that three of the authors of this paper are listed on as co-authors!) for the method used.

Line 309. Lateral flow? Suggest change to "lateral water export".

Line 372. Goodrich et al. (2015) were looking for a mechanism for elevated $CH_4$ fluxes occurring after rainfall events at a certain time of year, when the water table was within a narrow depth range. I

don't think rain events can be cited as a suppression cause based on this reference. Please check carefully.

Line 380-384. What about the annual flux reported by Goodrich et al. (2015)? 29 and 21 $gCH_4$-C m$^{-2}$yr$^{-1}$ (EC, not chamber).

Line 385 and Figure 7. I suggest that this figure be modified to show just summertime diurnal variation because: 1) annually-composed ensemble can mask season-specific diurnal variations; 2) apparently much of the wintertime data were missing (more needs to be said about this as noted above).

Section 4.5 title. Suggest replace "exchange" with "balance".

Line 410. Replace "'long-term" with "annual".

Line 412. "e.g. sustained step-change ….". I don't understand what this means. Please clarify.

Line 413. Please replace "were proposed" with "have been proposed".

Line 425. "Annual $CH_4$ emission was" (this was a single year study).

Line 430. Why exclude DOC flux? If it is because DOC flux was not measured in the subject year, then consider providing two NECB values, one without and one with estimated DOC.

Lines 433-436. I find this quite unsatisfying. "So what"? What useful message are we to take from this? A deeper philosophical discussion earlier would have helped, that questioned IPCC-type approaches as applied to restoration of long-term C sink ecosystems such as peatlands. What is the point of concluding with these diametrically opposing GWP estimates?

---

## Author Response (AR2)

[revised manuscript text omitted]

Sung-Ching Lee 2017-3-27 10:12 AM
格式設定: 字型色彩: 自動

Sung-Ching Lee 2017-3-27 10:12 AM
刪除: (Lehner and Döll, 2004; Mitsch et al., 2010)(Grasset et al., 2016; Mitsch et al., 2013)(Bortolotti et al., 2016; Lu et al., 2016; Petrescu et al., 2015)(Bridgham et al., 2006; Lal, 2008; Wisniewski and Sampson, 2012)(Bergamaschi et al., 2007; Bloom et al., 2010; Ciais et al., 2013)(Roulet, 2000; Yu et al., 2010)(Gorham, 1991; Menviel and Joos, 2012; Yu, 2011)(Frolking et al., 2013; Kleinen et al., 2010; Wania et al., 2009)Wetland ecosystems play a disproportionately large role in the global carbon (C) cycle compared to the surface area they occupy. Wetlands cover only 6%–7% of the Earth's surface (Lehner and Döll, 2004), but they act as a major sink for the long... [1]

Sung-Ching Lee 2017-3-27 10:12 AM
格式設定: 字型:(中文) Times New Roman, 英文 (英國), 使用拼字與文法檢查

Sung-Ching Lee 2017-3-27 10:12 AM
格式設定: 字型:(中文) Times New Roman, 英文 (英國), 使用拼字與文法檢查

Sung-Ching Lee 2017-3-27 10:12 AM
格式設定: 字型色彩: 自動

Sung-Ching Lee 2017-3-27 10:12 AM
格式設定: 字型:(中文) Times New Roman, 英文 (英國), 使用拼字與文法檢查

Sung-Ching Lee 2017-3-27 10:12 AM
格式設定: 字型:(中文) Times New Roman, 英文 (英國), 使用拼字與文法檢查

Sung-Ching Lee 2017-3-27 10:12 AM
格式設定: 字型:(中文) Times New Roman, 英文 (英國), 使用拼字與文法檢查

Sung-Ching Lee 2017-3-27 10:12 AM
格式設定: 字型:(中文) Times New Roman, 英文 (英國), 使用拼字與文法檢查

Sung-Ching Lee 2017-3-27 10:12 AM
格式設定: 字型:(中文) Times New Roman, 英文 (英國), 使用拼字與文法檢查

Sung-Ching Lee 2017-3-27 10:12 AM
格式設定: 字型:(中文) Times New Roman, 英文 (英國), 使用拼字與文法檢查

Sung-Ching Lee 2017-3-27 10:12 AM
格式設定: 字型:(中文) Times New Roman, 英文 (英國), 使用拼字與文法檢查

Sung-Ching Lee 2017-1-23 7:17 PM
格式設定: 字型色彩: 自動

Sung-Ching Lee 2017-1-23 7:17 PM

[revised manuscript text omitted]

Sung-Ching Lee 2017-3-2 8:48 PM
刪除: -…covariance system (EC-1) was open… [2]

Sung-Ching Lee 2017-3-2 8:56 PM
刪除: Fc

Sung-Ching Lee 2017-3-2 8:56 PM
格式設定 … [3]

Sung-Ching Lee 2017-1-23 7:43 PM
刪除: were calculated over 30 min blocks

Sung-Ching Lee 2017-3-27 10:13 AM
刪除: 2

Sung-Ching Lee 2017-3-27 10:14 AM
格式設定 … [5]

Unknown
功能變數代碼變更

Sung-Ching Lee 2017-1-23 7:46 PM
刪除: Gaps in climate data (<1% of a year) were filled using measurements at nearby climate stations as documented in Lee et al. (2016)

Sung-Ching Lee 2017-3-27 10:14 AM
格式設定 … [4]

Sung-Ching Lee 2017-3-27 10:14 AM
刪除: 4

Sung-Ching Lee 2017-3-27 10:28 AM
格式設定 … [6]

Nick 2017-1-31 5:21 PM
刪除: s

Sung-Ching Lee 2017-1-23 8:15 PM
格式設定: 字型色彩: 自動

**3.3.1 Gap filling of CO$_2$ flux data**

For gaps longer than 2 hours in CO$_2$ fluxes, the CO$_2$ flux (e.g., net ecosystem exchange, NEE) was modelled as the difference between ecosystem respiration ($R_e$) and gross ecosystem productivity (GEP) i.e. NEE = $R_e$ − GEP. Nocturnal NEE values were $R_e$ as there is no photosynthesis (GEP) at night.

$R_e$ was modelled based on soil temperature at the 5-cm depth ($T_{s,5cm}$) using a logistic fit (Neter et al., 1988):

$$R_e = \frac{1}{r_1 r_2^{T_{s,5cm}} + r_3} \qquad (1)$$

A comparable logistic function was proposed and used by FLUXNET Canada (Barr et al., 2002; Kljun et al., 2006). In this study, we used this logistic model available in IDL (version 8.5.1, Exelis Visual Information Solutions, Boulder, Colorado). $r_1$, $r_2$, and $r_3$ are empirical parameters; $r_1$ controls the slope of exponential phase; $r_2$ decides where the transitional phase starts; and $r_3$ determines the height of plateau phase. For each day of the year, the parameters $r_1$, $r_2$, and $r_3$ for $R_e$ were determined independently using a moving ± 60-day window centered on that day, based on all measured nighttime data from 2014 to 2016 when friction velocity was higher than 0.08 m s$^{-1}$. Lee (2016) determined the effect of using different window sizes (60, 90, 120 and full year) on the annual modelled and gap-filled $R_e$ and showed that a moving window size of 120 days was least sensitive to errors while still allowing for seasonal changes. However, sensitivity of choosing different window sizes on gap filled $R_e$ was small, varying the annual value between 226 and 245 g C m$^{-2}$ year$^{-1}$.

GEP was first partitioned from measured daytime NEE using modelled $R_e$. Any missing GEP data were then modelled using the photosynthetic light-response curves (Ögren and Evans, 1993) based on photosynthetic photon flux density (PPFD in µmol m$^{-2}$ s$^{-1}$):

$$\text{GEP} = \frac{MQY \cdot \text{PPFD} + P_M - ((MQY \cdot \text{PPFD} + P_M)^2 - 4 \cdot C_v \cdot MQY \cdot \text{PPFD} \cdot P_M)^{0.5}}{2 \cdot C_v} \qquad (2)$$

Maximum photosynthetic rate at light saturation ($P_M$) and maximum quantum yield ($MQY$) are fitted parameters with GEP estimated as measured daytime NEE minus daytime $R_e$ calculated using Eq. 1. Convexity ($C_v$) was fixed at 0.7 (Farquhar et al., 1980). For each day of the year, the time-varying parameters $MQY$ and $P_M$ were determined independently using a moving ± 45-day window centered on that day, using all data from 2014 to 2016 when friction velocity was higher than 0.08 m s$^{-1}$. The sensitivity of window size on gap filled GEP was small, resulting in annual value to vary between 385 and 415 g C m$^{-2}$ year$^{-1}$.

Sung-Ching Lee 2017-1-23 7:46 PM
删除: photosynthesis

Sung-Ching Lee 2017-3-27 10:16 AM
格式設定: 字型色彩: 自動
Sung-Ching Lee 2017-3-27 10:16 AM
格式設定: 字型色彩: 自動
Sung-Ching Lee 2017-3-27 10:16 AM
格式設定: 字型色彩: 自動
Sung-Ching Lee 2017-3-27 10:16 AM
格式設定: 字型色彩: 自動
Sung-Ching Lee 2017-3-27 10:16 AM
格式設定: 字型色彩: 自動
Sung-Ching Lee 2017-3-27 10:16 AM
格式設定: 字型色彩: 自動
Sung-Ching Lee 2017-3-27 10:16 AM
格式設定: 字型色彩: 自動
Sung-Ching Lee 2017-3-27 10:16 AM
格式設定: 字型色彩: 自動
Sung-Ching Lee 2017-3-27 10:16 AM
格式設定: 字型色彩: 自動
Sung-Ching Lee 2017-3-27 10:16 AM
格式設定: 字型色彩: 自動
Sung-Ching Lee 2017-3-27 10:16 AM
删除: The empirical parameters $r_1$, $r_2$, and $r_3$ were determined separately for each day of the year, using a moving window of 120 days (60 days into past and 60 days into future)
Sung-Ching Lee 2017-1-23 7:47 PM
删除: 221 and 229
Sung-Ching Lee 2017-3-27 10:17 AM
格式設定: 字型色彩: 自動
Sung-Ching Lee 2017-3-27 10:16 AM
删除: T
Sung-Ching Lee 2017-3-27 10:17 AM

[revised manuscript text omitted]

Sung-Ching Lee 2017-3-2 8:56 PM
刪除: F$_c$
Sung-Ching Lee 2017-3-2 8:56 PM
格式設定: 字型:非 斜體
Nick 2017-2-20 11:36 AM
格式設定: 字型:斜體
Sung-Ching Lee 2017-1-23 9:01 PM
格式設定: 字型:(中文) 新細明體, (中文) 中文 (台灣)

[Figure]

925 Figure 2: The annual course of weather variables ($T_a$, $T_s$, P, and PAR), ET, and WTH. The 30-year climate normals (30-year $T_a$ and P) were measured at Vancouver International Airport (Data: Environment Canada).

Sung-Ching Lee 2017-1-23 9:04 PM
格式設定: 下標
Sung-Ching Lee 2017-1-23 9:04 PM
格式設定: 下標

[Figure]

Figure 3: Monthly gap-filled $R_e$ (x-axis) drawn against GEP (y-axis). The resulting NEE can be read off the diagonal lines. The thick 1:1 line shows carbon neutrality, while lines in the upper right are of increasingly negative NEE (uptake) and lines towards the lower right are positive NEE (net source).

[Figure]

Figure 4: Isopleths of gap-filled NEE (net $CO_2$ fluxes) from the EC-1 system plotted as a composite in the study year. The graph uses a Gaussian filter of σ = 45 days (which conserves total NEE) to graphically smooth horizontal variations.

[Figure]

格式設定: 字型:(中文) +佈景主題王體王體亞洲
Cherry Chen 2017-3-29 5:19 PM

刪除:
Nick 2017-2-20 11:38 AM

[Figure]

940     Figure 5: Relationship between $R_e$ (nighttime 30-minute $CO_2$ flux measurements) and $T_{s,5cm}$ during the entire study period. The $u_*$ threshold was 0.08 m s$^{-1}$. The fitted curve is a logistic relationship following Eq. 1. $T_{s,5cm}$ was binned for 32 classes from minimum of $T_{s,5cm}$ to maximum of $T_{s,5cm}$. See Fig. S5 in supplement for seasonal differences. Negative $R_e$ values were caused by measurement uncertainties.

Sung-Ching Lee 2017-3-27 10:07 AM
删除: S3

[Figure]

Figure 6: Annual light response curve determined from the daytime 30-minute NEE measurements and Eq. 1, i.e., GEP = $R_e$ + -NEE. The curves are the best fit of the Eq. 2. PPFD was binned for 30 classes from 0 to 1500 µmol m$^{-2}$ s$^{-1}$. Annual *MQY* was 4.00 mmol C mol$^{-1}$ photons, $P_M$ was 4.68 umol m$^{-2}$ s$^{-1}$, and $C_v$ was 0.7 (fixed).

950

[Figure]

[Figure]

Figure 7: The ensemble diurnal course of measured CH$_4$ fluxes from the EC-2 system during the study period. More datasets were available from the summer half-year than from the winter half-year.

Sung-Ching Lee 2017-1-23 5:13 PM
删除:
Unknown
格式設定: 字型:(中文) 新細明體
Sung-Ching Lee 2017-3-27 7:04 PM
删除: (a)
Sung-Ching Lee 2017-3-27 10:28 AM
格式設定: 字型:(預設) +佈景主題標題
Sung-Ching Lee 2017-3-27 10:28 AM
格式設定: 字型:(預設) +佈景主題標題,
使用拼字與文法檢查
Sung-Ching Lee 2017-3-27 10:28 AM
格式設定: 字型:(預設) +佈景主題標題
Sung-Ching Lee 2017-3-27 10:28 AM
格式設定: 字型色彩: 自動
Sung-Ching Lee 2017-3-27 10:28 AM
格式設定: 字型色彩: 自動
Sung-Ching Lee 2017-1-23 7:04 PM
删除: Diurnal course of filled CH$_4$ fluxes from the
EC-2 system in the entire study period.
Sung-Ching Lee 2017-3-27 10:28 AM
格式設定: 字型:(預設) +佈景主題標題

[Figure]

960 Figure 8: EC-measured monthly $CO_2$, $CH_4$ and net GHGs fluxes shown as $CO_2e$ totals by using (a) 100-year and (b) 20-year GWPs. Missing data were gap-filled.

Table 1: Monthly EC-measured and gap-filled NEE ($CO_2$ fluxes), $CH_4$ fluxes, $CO_2$e fluxes using 20-year GWP, and $CO_2$e fluxes using 100-year GWP at the study site during the study period.

| Month | $R_e$ | GEP | NEE | $CH_4$ fluxes | 20-year $CO_2$e fluxes | 100-year $CO_2$e fluxes |
|---|---|---|---|---|---|---|
| | (g $CO_2$-C m$^{-2}$ month$^{-1}$) | | | (mg $CH_4$-C m$^{-2}$ month$^{-1}$) | (g $CO_2$e m$^{-2}$ month$^{-1}$) | (g $CO_2$e m$^{-2}$ month$^{-1}$) |
| Jan | 6.17 | 7.50 | -1.33 | 93 | 2.06 | -2.57 |
| Feb | 6.94 | 12.46 | -5.52 | 224 | -10.82 | -17.09 |
| Mar | 17.33 | 25.89 | -8.59 | 465 | -7.18 | -23.38 |
| Apr | 23.52 | 59.73 | -36.21 | 1170 | -35.33 | -100.29 |
| May | 36.46 | 92.63 | -56.20 | 1643 | -39.42 | -150.53 |
| Jun | 26.13 | 71.10 | -44.97 | 2670 | 144.23 | -61.85 |
| Jul | 38.53 | 61.47 | -22.94 | 4371 | 474.88 | 102.22 |
| Aug | 36.15 | 42.97 | -6.82 | 3813 | 492.32 | 147.44 |
| Sep | 24.84 | 25.08 | -0.21 | 1650 | 180.67 | 59.71 |
| Oct | 10.76 | 9.58 | 1.18 | 930 | 77.23 | 28.62 |
| Nov | 5.16 | 3.39 | 1.77 | 240 | 19.93 | 10.97 |
| Dec | 3.63 | 2.79 | 0.87 | 155 | 10.13 | 5.50 |
| Study year | g $CO_2$-C m$^{-2}$ year$^{-1}$ | | | g $CH_4$-C m$^{-2}$ year$^{-1}$ | g $CO_2$e m$^{-2}$ year$^{-1}$ | |
| | 236 ± 16.4 | 415 ± 28.8 | -179 ± 26.2 | 17 ± 1 | 1248 ± 147.6 | -22 ± 103.1 |

965

Table 2: Comparison of annual NEE, $R_e$ and GEP, over different ecosystems (vegetation covers) in the Vancouver region using EC measurements. Sorted by magnitude of -NEE/GEP ratio.

| Site | Land cover | NEE | $R_e$ | GEP | -NEE/GEP |
|---|---|---|---|---|---|
| | | g C m$^{-2}$ year$^{-1}$ | | | |
| **Burns Bog** **(this study)** Delta, BC | Rewetted raised bog ecosystem | -179 | 236 | 415 | 43% |
| **Westham Island** (CA-Wes)[*] Delta, BC | Unmanaged grassland | -222 | 1215 | 1438 | 15% |
| **Campbell River** (CA-Ca1)[*] Vancouver Island | Douglas-fir forest (~55 yrs) | -328[+] | 1830[+] | 2158[+] | 15% |
| **Buckley Bay** (CA-Ca3)[*] Vancouver Island | Douglas-fir forest (~15 yrs) | 64[+] | 1487[+] | 1423[+] | -4% |

1050   [*] Site identifier in global FLUXNET database (http://fluxnet.ornl.gov).
[+] Data from Krishnan et al., 2009 before fertilisation.

1055

**MAIN COMMENTS TO THE AUTHOR(S)**

**1) There has been no attempt to discuss or quantify the likely size of uncertainties of the fluxes, especially as they relate to annual fluxes and CO2-e values. It is not really acceptable to publish these data now without considering the**

1060 **likely size of uncertainties. Given my comments below about gap filling CH4 fluxes, I strongly suggest that a due consideration of the magnitude of uncertainties at the annual timescale is warranted.**

[Response]

As suggested, we have estimated uncertainties and added a new section (Section 3.3.3) describing the method we used to estimate the uncertainties, followed by including related text in the Results and Discussion as follows:

1065

"3.3.3 Error estimates

The uncertainty associated with annual estimates of NEE, GEP, $R_e$ and $CH_4$ fluxes resulting from gap filling and due to different window sizes was quantified as follows: First, in the annual dataset of half-hourly fluxes random gaps were inserted using Monte Carlo simulation (Griffis et al., 2003; Krishnan et al., 2006; Paul-Limoges et al., 2015); The maximum number

1070 of gaps were set to 40 and the maximum length was set to 10 days resulting in total gaps of on average 28% of the year (and up to 40% of the year). The Monte Carlo simulation was run 500 times and the 95% confidence intervals were used to calculate the uncertainty of the annual sums.

Secondly, the uncertainty associated with choosing different window sizes for the derivation of the relationships in the gap-filling (see Section 3.3.1 and 3.3.2) was estimated from a range of annual values obtained using window sizes of 30, 45,

1075 60, 75, 90, 120, 150, 180, and 365 days for GEP, $R_e$, and NEE; the same selections of window sizes with three additions (210, 240, and 270 days) were applied for calculating the uncertainty of the annual $CH_4$ budget. The overall uncertainty in the annual estimates of NEE, GEP, $R_e$ and $CH_4$ fluxes was then obtained by taking the square root of the sum of squares of the error from the gap filling (Monte Carlo simulation) and the uncertainty of the estimates due to different window sizes."

1080 The average and uncertainty values for annual NEE, $R_e$, GEP, and $CH_4$ budget were -179 $\pm$ 26.2, 236 $\pm$ 16.4, 415 $\pm$ 28.8, and 17 $\pm$ 1.0 g C m$^{-2}$ year$^{-1}$, respectively. All numbers in the abstract and manuscript, including $CO_2$e estimates (1248 $\pm$ 147.6 g$CO_2$e m$^{-2}$ year$^{-1}$ for the 20-year time horizon and -22 $\pm$ 103.1 g$CO_2$e m$^{-2}$ year$^{-1}$ for the 100-year time horizon), have been updated accordingly.

1085 **2) My comments about gap filling CH4 fluxes have not really been adequately addressed. I suggest that the nature of the relationship settled on in 3.3.2 be supported by a figure, even if it is in supplementary material. I do find it surprising that an r2 as high as 66% was found, unless it is simply the result of the extremely wide range of flux values. If that were the case, I'd be more concerned about r2 within the region of the relationship where most of the annual flux was derived (i.e. during warm temperatures). This is also why an estimate of uncertainty is required.**

1090 **Moreover, the authors hinted that little CH4 flux data were available in winter, increasing uncertainty. They also hint (lines 328-329) that there may well have been hysteresis in the temperature vs Fm relationship. It may well be more defensible to gap fill using the means of daily 30-minute measurements to construct daily means. But without seeing the data, it's difficult to judge. Suggest looking at Goodrich et al. (2015) JGR-BG (DOI: 10.1002/2014JG002844) for a situation with strong hysteresis in methane fluxes from a wetland, where a "simple" regression-based approach was**

1095 **demonstrated to be inadequate.**

[Response]

We have improved our analysis with the aim of determining the contributions of additional environmental controls (different depths of soil temperature, air temperature, water table depth, soil water content, and NEE). We have described the additional analyses in the text, and have added two figures and two tables to provide additional information.

1100

We have rewritten Section 3.3.2 as follows:

"CH$_4$ fluxes with quality flags 0 and 1 according to Mauder and Foken (2004) were plotted against all relevant variables including NEE, *WTH*, $\theta_w$, $T_a$, $T_{s,5cm}$, $T_{s,10cm}$, and $T_{s,50cm}$. The highest correlation between a single variable and the CH$_4$ flux

1105 was found for soil temperature using an exponential relationship (Fig. S3). Of the soil temperatures measured at three different depths, $T_{s,10cm}$ explained the highest proportion of the variance in CH$_4$ flux (Table S1). Therefore, $T_{s,10cm}$ was used to build an initial model and a logarithmic transformation of the CH$_4$ fluxes was applied to remove the heteroscedasticity and permit the use of a linear regression model. Then the residual analysis was applied to explore whether the variance in the residual could be explained by other controls. The residual was defined as the ratio of the measured CH$_4$ fluxes to the

1110 modelled CH$_4$ fluxes from the initial model. Based on the residual analysis, the main contributor to the residual, *WTH*, explained 7% of the variance (Table S2). Additionally, there was a hysteresis relationship between CH$_4$ flux and *WTH* (Fig. S4). In order to have a more robust gap filling model, $T_{s,10cm}$ and *WTH* were used to fill the gaps in CH$_4$ fluxes. We used a combination of an exponential temperature response function and a linear *WTH* function as follows:

1115
$$F_m = (aWTH + b)e^{cT_{s,10cm}} \qquad (3)$$

where $a, b$, and $c$ are time-varying empirical parameters. The three parameters were fitted separately for each day, using a moving window of ±105 days using all data from the study period when friction velocity was greater than 0.08 m s$^{-1}$. Overall, 76% of the variance of the CH$_4$ fluxes was explained by $T_{s,10cm}$ and $WTH$. The combination of soil temperature and $WTH$ has also been shown to explain a large proportion of the observed variances in CH$_4$ fluxes in peatlands in other studies (Brown et al., 2014; Goodrich et al., 2015)."

Second, two new figures (Fig. S3 and S4) and two tables (Table S1 and S2) have been added in supplementary material:

[Figure]

1125    Figure S3. The relationships between CH₄ fluxes with quality flags 0 and 1 and all measured environmental factors using all

half-hourly data.

[Figure]

[Figure]

Figure S4. (a) Monthly mean *WTH* vs. monthly mean $T_{s,10cm}$. (b) Monthly mean $CH_4$ flux based on available measurements (non-gap-filled) vs. monthly mean *WTH* and (c) $T_{s,10cm}$.

Table S1. Coefficients of determination ($R^2$) from simple regression analyses of relationships between $CH_4$ fluxes and soil temperature ($T_a$, $T_{s,5cm}$, $T_{s,10cm}$, $T_{s,50cm}$, $\theta_W$, WTH and NEE. The highest value was marked as orange.

| Initial identification of major controls | | | | | | | |
|---|---|---|---|---|---|---|---|
| Variables / Functions | $T_{s,5cm}$ | $T_{s,10cm}$ | $T_{s,50cm}$ | $T_a$ | $\theta_W$ | WTH | NEE |
| Linear | 0.55 | 0.59 | 0.48 | 0.21 | 0.57 | 0.55 | 0.00 |
| Exponential | 0.66 | **0.73** | 0.64 | 0.23 | 0.51 | 0.51 | 0.01 |
| Logarithmic | 0.50 | 0.54 | 0.48 | 0.28 | 0.50 | 0.00 | 0.00 |
| Polynomial | 0.56 | 0.61 | 0.48 | 0.31 | 0.57 | 0.52 | 0.01 |

1135    Table S2. The residual analysis for $T_a$, $\theta_W$, WTH and NEE for the study period based on the relationship between $CH_4$ flux $T_{s,10cm}$. The highest value was marked as orange.

| | NEE | WTH | $\theta_W$ | $T_a$ |
|---|---|---|---|---|
| $R^2$ value | 0.004 | **0.065** | 0.045 | 0.022 |

Finally, the monthly $CH_4$ fluxes and annual $CH_4$ budget were re-calculated using the soil temperature at 10 cm compared to
1140    5 cm in the discussion paper. This resulted in a change of the annual $CH_4$ flux from 16 to 17 g $CH_4$-C m$^{-2}$ year$^{-1}$. Also $CO_2$e fluxes were recalculated. The corresponding text has been updated (Line 32, 360, 362, and 393).

**SPECIFIC COMMENTS TO THE AUTHOR(S)**

l145 **1) Throughout: "eddy-covariance" should be "eddy covariance". The hyphen is unnecessary.**

[Response]

This has been changed as proposed.

**2) Line 1 of abstract: peatlands have been drained for many other purposes that have also changed them into C**
l150 **sources.**

[Response]

This sentence has been edited as follows:

"Many peatlands have been drained and harvested for peat mining, agriculture, and other purposes, which has turned them from carbon (C) sinks into C emitters."

l155

We also changed the text on line 50:

"Many peatlands have been harvested and continue to be disturbed by the extraction of peat for horticultural use and conversion to agriculture as well as other purposes."

l160 **3) Introduction and lines 38-40. I suggest that the authors might consider constraining their introduction (or at least detailed examples of fluxes) to peatlands. Wetlands are so incredibly diverse that it is almost meaningless to provide context for a study focused on a restored peatland. The Grasset et al. (2016) paper seems a rather strange choice because this appears to have been an aquatic study and not even focussed on annual fluxes. I couldn't find the CO2-C values cited in this or the Mitsch et al. (2013) papers. It is really important at this stage of a manuscript that the**
l165 **authors concentrate on verifying their source material and providing relevant and authoritative literature sources.**

[Response]

We have followed the reviewer's advice and focused on peatlands. We agree selected the wetlands were not comparable. The first paragraph of introduction has been re-written as follows:

l170 "Wetland ecosystems play a disproportionately large role in the global carbon (C) cycle compared to the surface area they occupy. Wetlands cover only 6% – 7% of the Earth's surface (Lehner and Döll, 2004; Mitsch et al., 2010) but C storage in wetlands has been estimated to be up to 450 Gt C or approximately 20% of the total C storage in the terrestrial biosphere (Bridgham et al., 2006; Lal, 2008; Wisniewski and Sampson, 2012). On the other hand, they emit significant quantities of methane ($CH_4$), a powerful greenhouse gas (GHG), which is responsible for 30% of all global $CH_4$ emissions (Bergamaschi

l175 et al., 2007; Bloom et al., 2010; Ciais et al., 2013) due to anaerobic microbial decomposition (Aurela et al., 2001; Rinne et

al., 2007). Peatlands are the most widespread of all wetland types in the world representing 50 to 70% of global wetlands (Roulet, 2000; Yu et al., 2010. Peatlands around the world sequester around 50 g $CO_2$-C $m^{-2}$ $year^{-1}$ (Roulet et al., 2007; Christensen et al., 2012; Humphreys et al., 2014; McVeigh et al., 2014; Peichl et al., 2014, Pelletier et al., 2015) and emit around 12 g $CH_4$-C $m^{-2}$ $year^{-1}$ (Abdalla et al., 2016; Brown et al., 2014; Jackowicz-Korczynski et al., 2010; Lai et al., 2014; Urbanova et al., 2013). Futhermore, it has been shown that it is crucial to include peatlands in the modelling and analysis of the global C cycle (Frolking et al., 2013; Kleinen et al., 2010; Wania et al., 2009)."

**4) Line 73 onward. Using the term "long-term" suggests multi-year. Clearly this paper is not multi-year so, at least in the conclusion, the authors might need to acknowledge the additional uncertainties affecting their conclusions caused by there being a single year of study.**

[Response]

We appreciate reviewer's comments. We have changed this to "year-round" for clarity.

**5) Lines 130-131. The change to the 60 Hz detail has not been made.**

[Response]

We appreciate the reviewer noticing our slip here. The change has now been made.

**6) Line 146. "Small gaps ... and H2O fluxes"?**

[Response]

This information has been added as follows:

"Small gaps (<60 minutes) of missing $CO_2$, $H_2O$, and $CH_4$ fluxes were filled by linear interpolation."

The gap filling method for larger gaps in $H_2O$ fluxes has also been added as mentioned in our response to Comment #27.

**7) Line 154 "atmosphere" is mis-spelled.**

[Response]

We have corrected the spelling error.

**8) Use of terms Fc and NEE. In general there is inconsistent use of these terms which, to an unfamiliar reader, might be very confusing. For example, why on line 155 use "NEE and Fm" rather than "Fc and Fm"?**

[Response]

We appreciate reviewer's suggestion. We decided to delete $F_c$ in the manuscript and figure captions for clarity.

**9) Lines 167-168. It is not really accurate to state that the parameters "were determined separately for each day of the year", when the dataset for each day consisted of a window of 120 days width! Consider a more accurate description.**

[Response]

We appreciate reviewer's comment. This is a misunderstanding, the moving window was 120 days for $R_e$, but for each day the parameters were determined independently using a $\pm$ 60-day moving window centered on that day. We have edited the text for clarity at line 168 and 181 as follows:

"… the height of plateau phase. For each day of the year, the parameters $r_1$, $r_2$, and $r_3$ for $R_e$ were determined independently using a moving $\pm$ 60-day window centered on that day based on all …"

"… was fixed at 0.7 (Farquhar et al., 1980). For each day of the year, the time-varying parameters $MQY$ and $P_M$ were determined independently using a moving $\pm$ 45-day window centered on that day using all data from 2014 to 2016 when friction velocity was …"

**10) Lines 173-182. This does not actually describe completely how GEP was modelled. The first step, surely, was to partition GEP out from measured values of NEE (using modelled Re), and then the light response function was fitted.**

[Response]

We appreciate reviewer's comments. This information has been added as follows (at line 174):

"GEP was first partitioned from measured daytime NEE using modelled $R_e$. Any missing GEP data were then modelled using the photosynthetic light-response curves (Ögren and Evans, 1993) based on …"

**11) Line 207. Include reference to (the new) Fig. 2.**

[Response]

This has been implemented as follows (at line 210):

"The study site received a total annual precipitation of 1062 mm, of which 16% (174 mm) fell during the warm half year (Apr-Sep) and 84% (888 mm) during the cold half year (Oct-Mar) (Fig. 2)."

**12) Line 246. "Despite...". It is not quite clear what this sentence means. Maybe replace with "Compared to ...".**

[Response]

This has been implemented as follows:

"Despite Compared to a large seasonal amplitude in monthly GEP, $R_e$ showed less variability over the year."

**13) Line 247. This should be a reference to Fig. 3, not Fig. 4.**

[Response]

Thanks for pointing out the error. We have corrected the reference.

**14) Line 264. Replace "pristine" with "more or less pristine". For example, the Campbell et al. (2014) study was for a drainage impacted peatland. It is hard to judge what "pristine" actually means, and so most wetland scientists will avoid the term.**

[Response]

We appreciate reviewer's suggestion. The word, "undisturbed", has been implemented to avoid the controversy as follows:

"Values that are comparable to the current restored wetland were reported in five undisturbed temperate wetlands: …"

**15) Lines 277-281. There needs to be some details provided about the methodology used to calculate DOC fluxes, since the reference is unpublished. At least it is nice to see some uncertainty value provided!**

[Response]

We appreciate reviewer's comment. This work is not yet peer-reviewed, but has been published and is accessible though AGU 2016 (https://agu.confex.com/agu/fm16/meetingapp.cgi/Paper/165988) where the complete poster contribution describing the method is also available. To provide some context though, we have added some information regarding to the methodology used to measure DOC fluxes as follows (at line 282):

"Estimation of DOC fluxes was based on regular (approx. monthly) water samples collected at 5 locations within the flux tower footprint using the headspace equilibration technique. Lateral flow was estimated as the residual of the water balance."

**16) Line 284. Where is the "daily maximum in GEP" shown?**

[Response]

This specific number was calculated. It is not shown in the figure. Please note that the full dataset of this study will be made available at the time of publication in a data repository.

**17) Lines 286-287. "The rapid decrease in monthly Re (Figure 3?) from May to June ...". Fig. 3 does not support this statement.**

[Response]

We appreciate reviewer's suggestion. Figure 3 shows this change too, but in order to be clear, a reference to Table 1 which best shows this decrease has been added as follows:

"The rapid decrease in monthly $R_e$ from May to June (Table 1) was caused by …"

**18) Lines 319-321. How were these controls investigated? How did you determine "the effects of Ta on GEP were approximately limited between 10 and 15 C."? (The underlined words make no sense!) It is notoriously difficult to isolate the effects of temperature versus light quality versus VPD, because they are all strongly correlated.**

[Response]

We appreciate reviewer's comments. These two variables (WTH and $T_a$) were plotted against gap-filled GEP to let us investigate their effects on NEE. To clarify this, we rewrote this short paragraph as follows:

"Other possible controls on GEP explored were WTH and $T_a$. We found that WTH was not a control on GEP (R$^2$ = 0.08) in the current study as the study area remained fairly wet throughout the year. Furthermore, the effect of $T_a$ on GEP was less and limited to a smaller temperature range, compared to $T_s$. approximately limited between 10 and 15 °C."

**19) Line 326. Still using the inaccurate term "significant". Do you mean "Much larger"?**

[Response]

We appreciate reviewer's suggestion and apologize for not making the change. The change has been made as follows:

"…, and then became a much larger significant source in summer …"

**20) Lines 328-329. Hysteresis in the Ta vs Fm relationship is hinted at. This suggests that the simplistic gap filling method might be quite inaccurate at these times.**

[Response]

The reviewer has made an important point. In order to explore a possible hysteresis effect, we analyzed the dataset further by plotting the monthly (not gap filled) CH$_4$ flux vs. $T_{s,10cm}$ and also WTH (Fig. S2, or please refer to our response to Comment #2 in the major comment section). We also graphed monthly $T_{s,10cm}$ vs. WTH which shows that a hysteresis effect is observed, as WTH and $T_{s,10cm}$ are shifted in phase. Based on these additional analyses, we have updated the CH$_4$ gap filling model to include both, $T_{s,10cm}$ and WTH, and added a short paragraph to discuss the possible reasons for the seasonality of CH$_4$ fluxes (see our answer to Comment #25 below). Thus we have deleted the sentence under reference to avoid any misunderstanding.

"CH$_4$ fluxes showed a seasonal pattern, which was linked to phenology and temperature."

**21) Lines 346-347. Why is this surprising when it has been reported by many other studies?**

[Response]

We appreciate reviewer's question. This part has been re-written for clarity as follows:

"Surprisingly, there was only a small diurnal variation observed for CH$_4$ fluxes in the summer months, while larger diurnal variations have been found in other studies (Juutinen et al., 2004; Long et al., 2010; Sun et al., 2013; Wang and Han, 2005)."

**22) Line 365. "... magnitude of fluxes changed differently ..." is fairly meaningless. What do you mean?**

[Response]

We appreciate the comment. As we explained in the previous lines (362 – 365), the early onset of $CO_2$ sequestration caused the most negative net GHG forcing from April to June; the subsequent rapid drop in $CO_2$ sequestration and lagged $CH_4$ fluxes led to the highest net GHG forcing in July and August.

**23) Lines 366-367. Why should they be expected to be representative?**

[Response]

We appreciate reviewer's point. If the climate is steady or the effects of environmental variables are small, measuring the fluxes in the growing season, which is the time with least gaps due to sufficient solar charging, would be sufficient to provide a reasonable estimate of the annual budget. Moreover, we want to emphasize that a short-term campaign can be a good way to identify important site processes but the determination of the annual budget requires reliable long-term measurements. Thus, we have edited the text from Line 366 to 367 for clarity as follows:

"The results show that measurements made during a part of the growing season are not necessarily representative for the entire growing season or the year; a short-term campaign can be a good way to identify important site processes but the determination of the annual budget requires reliable long-term measurements."

**24) Lines 368-371. This paragraph is a very weak response to Reviewer 1's comment. It is barely coherent. It does not address the long-term nature of C accumulation (from CO2) in wetlands vs the short lifetime of CH4 in the atmosphere.**

[Response]

We appreciate reviewer's comments. We have addressed the point by pointing out that actual radiative forcing from sustained $CO_2$ and $CH_4$ fluxes are not included in GWP in the last version.

**25) Line 383. It is surely unusual that there is such a huge seasonal variation in Fm, especially given the temperate climate and high water table in winter. Can the authors suggest why this might be the case? The proper place for this is in Section 4.4.1.**

[Response]

We appreciate reviewer's comments. We have added a short paragraph at the end of the first paragraph in Section 4.4.1 which discusses the possible reasons for this as follows:

"Although it has been suggested that in some peatlands, WTH acts as a main control on $CH_4$ fluxes (Drösler et al., 2008; Knorr et al., 2009; Romanowicz et al., 1995; Roulet et al., 1993; Windsor et al., 1992), it has also been found that $CH_4$

emissions from wet soils (where the water table fluctuates within a small range near the surface) are highly dependent on $T_s$

1340 because the oxidation in a shallow top soil is negligible (Jackowicz-Korczynski et al., 2010; Long et al., 2010; Olson et al., 2013; Rinne et al., 2007; Song et al., 2009). In our study, $CH_4$ emissions in the summer months were relative high even when the water table dropped to around 20 cm below the surface, likely because the peat maintained anaerobic conditions above the water table (as discussed in Hendriks et al., 2007). In addition, one needs to consider the transport pathways for $CH_4$ which may help explain the higher $CH_4$ fluxes in summer. First, the presence of sedges created an effective additional

1345 diffusion pathway for $CH_4$ through the plants' aerenchyma (Herbst et al., 2011; Treat et al., 2007). Second, a high water table especially when it rises above the soil surface increases the diffusion resistance to $CH_4$ transport (Brown et al., 2014; Walter and Heimann, 2000). An additional reason for the lowest $CH_4$ emissions occurring in winter months could be the suppression from rain events (Goodrich et al., 2015) as 84% of rain fell at the site during the cold half-year."

1350 **26) Line 387. If the flux of DOC is taken into account, the annual C sink strength is less than this.**

[Response]

We appreciate reviewer's comments. We have re-written this sentence for clarity as follows:

"In terms of the C balance (DOC fluxes excluded), our results suggest that our study area in BBECA was a net C sink …"

1355 **27) Figure 2. Caption does not mention ET. Was it measured at the site and gapfilled or is it estimated from climate station values?**

[Response]

We appreciate reviewer's question. ET has been added to the caption, and the text describing the gap filling method for $H_2O$ fluxes has been added in Section 3.3 on line 149 as follows:

1360

"Longer gaps in $H_2O$ fluxes were filled with the online tool developed by the Max Planck Institute for Biogeochemistry in Jena, Germany. This tool uses the look-up table method documented in Falge et al. (2001) and Reichstein et al. (2005)."

**28) Figure 3 caption. The x-axis label (F_CO2) has not been corrected.**

1365 [Response]

We appreciate the reviewer noticing this. The version of figure was incorrect, it has been replaced by the correct one.

**29) Figure 7 caption is inadequate. Insert "Ensemble". Note that mainly summer is represented.**

[Response]

1370 We appreciate reviewer's suggestion. The information has been added as follows:

"The ensemble diurnal course of measured $CH_4$ fluxes from the EC-2 system during the study period. More datasets were available from the summer half-year than from the winter half-year."

**30) Figure S3. It would be preferable to use a common x-axis scale on all of these panels (as was done for Fig S4).**

l375 [Response]

We appreciate reviewer's comments. This change has been implemented as follows:

[Figure]

**Soil temperature (°C)**

1380 **31) Figures S3 and S4. It is still a little confusing what "on the first day of each time period" means. Presumably this is the function derived for the first day of the period – which is actually derived from data sourced 60 days prior to 60 days after. Surely it would make more sense to choose a day close to the center of the time period.**

[Response]

We appreciate reviewer's comments. This subtitle of each figure has been edited for clarity as follows:

[Figure]

1385

---

## Author Response (AR3)

[revised manuscript text omitted]

Sung-Ching Lee 2017-1-23 8:42 PM

Sung-Ching Lee 2017-1-23 8:42 PM

Sung-Ching Lee 2017-1-23 8:42 PM

Sung-Ching Lee 2017-1-23 8:42 PM

Sung-Ching Lee 2017-1-23 8:43 PM

Sung-Ching Lee 2017-1-23 8:43 PM

Sung-Ching Lee 2017-1-23 7:51 PM

Sung-Ching Lee 2017-1-23 8:43 PM

Sung-Ching Lee 2017-1-23 9:07 PM

Sung-Ching Lee 2017-3-27 10:07 AM

Sung-Ching Lee 2017-1-23 8:43 PM

Sung-Ching Lee 2017-1-23 4:50 PM

Sung-Ching Lee 2017-1-23 8:43 PM

Sung-Ching Lee 2017-1-23 4:50 PM

Sung-Ching Lee 2017-1-23 4:50 PM

**4.3.4 Gross ecosystem productivity**

Figure 6 shows the average light response curve, with half-hourly GEP as a function of PPFD. Due to different phenology over the year and the changes in solar altitude, light response curves were also calculated every two months (see supplementary material, Fig. S6). GEP reached a maximum in May with 92.63 g C m$^{-2}$ month$^{-1}$, and a minimum of 2.79 g C m$^{-2}$ month$^{-1}$ in December (Fig. 3, Table 1). GEP at light saturation reached roughly 5.09 μmol m$^{-2}$ s$^{-1}$ in summer, and remained below 2.49 μmol m$^{-2}$ s$^{-1}$ in winter, due to reduced leaf area, flooding, and lower temperatures. From March to May, GEP increased much more rapidly than $R_e$. In fall, GEP decreased faster than $R_e$. The magnitude of $R_e$ already was close to GEP in the late August to make the study area become $CO_2$ neutral in late summer.

Other possible controls on GEP explored were WTH and $T_a$. We found that WTH was not a control on GEP ($R^2$ = 0.08) in the current study as the study area remained fairly wet throughout the year. Furthermore, the effect of $T_a$ on GEP was less and limited to a smaller temperature range, compared to $T_s$.

**4.4 CH₄ exchange**

**4.4.1 Annual and seasonal CH₄ budgets**

Overall, the study area was a source of $CH_4$ in each of the twelve months (Table 1). The annual $CH_4$-C budget was 17 ± 1.0 g $CH_4$-C m$^{-2}$ yr$^{-1}$. $CH_4$ emissions were close to zero in winter (5.2 mg $CH_4$-C m$^{-2}$ day$^{-1}$). Seasonally, it was a weaker $CH_4$ source in fall (31.3 mg $CH_4$-C m$^{-2}$ day$^{-1}$) and spring (36.4 mg $CH_4$-C m$^{-2}$ day$^{-1}$), and then became a much larger source in summer (126.0 mg $CH_4$-C m$^{-2}$ day$^{-1}$). Monthly emissions of $CH_4$ ranged from 93 (January) to 4371 (July) mg $CH_4$-C m$^{-2}$ month$^{-1}$. The rising $T_a$ did not trigger $CH_4$ production immediately, and $CH_4$ fluxes remained low in April and May. But once the subsurface and water became warm enough, $CH_4$ emissions increased from to 1.4 to 2.7 g $CH_4$-C m$^{-2}$ month$^{-1}$ in June (Table 1). $CH_4$ emissions reached the peak in July (4.4 g $CH_4$-C m$^{-2}$ month$^{-1}$) and held similar magnitude (3.8 g $CH_4$-C m$^{-2}$ month$^{-1}$) in August even though the $T_a$ had dropped. Although it has been suggested that in some peatlands, WTH acts as a main control on $CH_4$ fluxes (Drösler et al., 2008; Knorr et al., 2009; Romanowicz et al., 1995; Roulet et al., 1993; Windsor et al., 1992), it has also been found that $CH_4$ emissions from wet soils (where the water table fluctuates within a small range near the surface) are highly dependent on $T_s$ because the oxidation in a shallow top soil is negligible (Jackowicz-Korczynski et al., 2010; Long et al., 2010; Olson et al., 2013; Rinne et al., 2007; Song et al., 2009). In our study, $CH_4$ emissions in the summer months were relative high even when the water table dropped to around 20 cm below the surface, likely because the peat maintained anaerobic conditions above the water table (as discussed in Hendriks et al., 2007). In addition, one needs to consider the transport pathways for $CH_4$ which may help explain the higher $CH_4$ fluxes in summer. First, the presence of sedges created an effective additional diffusion pathway for $CH_4$ through the plants' aerenchyma (Herbst et al., 2011; Treat et al., 2007). Second, a high water table especially when it rises above the soil surface increases the diffusion resistance to $CH_4$ transport (Brown et al., 2014; Walter and Heimann, 2000).

Sung-Ching Lee 2017-1-23 7:46 PM

Sung-Ching Lee 2017-1-23 9:07 PM

Sung-Ching Lee 2017-3-27 10:21 AM
**Formatted** … [28]

Sung-Ching Lee 2017-1-23 7:52 PM

Sung-Ching Lee 2017-3-27 10:00 AM

Sung-Ching Lee 2017-3-29 4:56 PM
**Formatted** … [30]

[revised manuscript text omitted]

Sung-Ching Lee 2017-1-23 5:13 PM

Sung-Ching Lee 2017-1-23 7:04 PM

Sung-Ching Lee 2017-3-27 10:28 AM

Sung-Ching Lee 2017-1-23 7:04 PM

Sung-Ching Lee 2017-3-27 10:28 AM

Sung-Ching Lee 2017-3-27 10:28 AM

Sung-Ching Lee 2017-3-27 10:28 AM

Sung-Ching Lee 2017-3-27 10:28 AM

Sung-Ching Lee 2017-3-27 10:28 AM

Sung-Ching Lee 2017-3-27 10:28 AM

[Figure]

Figure 8: EC-measured monthly $CO_2$, $CH_4$ and net GHGs fluxes shown as $CO_2e$ totals by using (a) 100-year and (b) 20-year GWPs. Missing data were gap-filled.

Table 1: Monthly EC-measured and gap-filled NEE ($CO_2$ fluxes), $CH_4$ fluxes, $CO_2$e fluxes using 20-year GWP, and $CO_2$e fluxes using 100-year GWP at the study site during the study period.

| Month | $R_e$ | GEP | NEE | $CH_4$ fluxes | 20-year $CO_2$e fluxes | 100-year $CO_2$e fluxes |
|---|---|---|---|---|---|---|
| | (g $CO_2$-C m$^{-2}$ month$^{-1}$) | | | (mg $CH_4$-C m$^{-2}$ month$^{-1}$) | (g $CO_2$e m$^{-2}$ month$^{-1}$) | (g $CO_2$e m$^{-2}$ month$^{-1}$) |
| Jan | 6.17 | 7.50 | -1.33 | 93 | 2.06 | -2.57 |
| Feb | 6.94 | 12.46 | -5.52 | 224 | -10.82 | -17.09 |
| Mar | 17.33 | 25.89 | -8.59 | 465 | -7.18 | -23.38 |
| Apr | 23.52 | 59.73 | -36.21 | 1170 | -35.33 | -100.29 |
| May | 36.46 | 92.63 | -56.20 | 1643 | -39.42 | -150.53 |
| Jun | 26.13 | 71.10 | -44.97 | 2670 | 144.23 | -61.85 |
| Jul | 38.53 | 61.47 | -22.94 | 4371 | 474.88 | 102.22 |
| Aug | 36.15 | 42.97 | -6.82 | 3813 | 492.32 | 147.44 |
| Sep | 24.84 | 25.08 | -0.21 | 1650 | 180.67 | 59.71 |
| Oct | 10.76 | 9.58 | 1.18 | 930 | 77.23 | 28.62 |
| Nov | 5.16 | 3.39 | 1.77 | 240 | 19.93 | 10.97 |
| Dec | 3.63 | 2.79 | 0.87 | 155 | 10.13 | 5.50 |
| Study year | g $CO_2$-C m$^{-2}$ year$^{-1}$ | | | g $CH_4$-C m$^{-2}$ year$^{-1}$ | g $CO_2$e m$^{-2}$ year$^{-1}$ | |
| | 236 ± 16.4 | 415 ± 28.8 | -179 ± 26.2 | 17 ± 1 | 1248 ± 147.6 | 22 ± 103.1 |

Formatted ... [38]
Sung-Ching Lee 2017-3-26 3:49 PM
Formatted ... [40]
Sung-Ching Lee 2017-3-26 3:49 PM
Formatted ... [41]
Sung-Ching Lee 2017-3-26 3:49 PM
Formatted ... [42]
Nick 2017-3-8 1:54 PM
Formatted ... [39]
Sung-Ching Lee 2017-3-26 3:49 PM
Formatted ... [43]
Sung-Ching Lee 2017-3-26 3:49 PM
Formatted ... [45]
Sung-Ching Lee 2017-3-26 3:49 PM
Formatted ... [47]
Sung-Ching Lee 2017-3-26 3:49 PM
Formatted ... [34]
Sung-Ching Lee 2017-3-26 3:49 PM
Formatted ... [37]
Nick 2017-3-8 1:54 PM
Formatted ... [35]
Sung-Ching Lee 2017-3-26 3:49 PM
Formatted Table ... [36]
Sung-Ching Lee 2017-3-26 3:49 PM
Formatted ... [44]
Sung-Ching Lee 2017-3-26 3:49 PM
Formatted ... [46]
Sung-Ching Lee 2017-3-26 3:49 PM
Formatted ... [48]
Sung-Ching Lee 2017-3-26 3:49 PM
Formatted ... [49]
Nick 2017-3-8 1:54 PM
Formatted ... [50]
Sung-Ching Lee 2017-3-26 3:49 PM
Formatted ... [51]
Nick 2017-3-3 2:59 PM
Nick 2017-3-3 2:59 PM
Sung-Ching Lee 2017-3-26 3:49 PM
Formatted ... [52]
Sung-Ching Lee 2017-3-26 3:49 PM
Formatted ... [66]
Nick 2017-3-8 1:54 PM
Formatted ... [53]
Sung-Ching Lee 2017-3-26 3:49 PM
Formatted ... [54]
Sung-Ching Lee 2017-3-26 3:49 PM
Formatted ... [55]
Sung-Ching Lee 2017-3-26 3:49 PM
Formatted ... [57]
Nick 2017-3-8 1:54 PM
Formatted ... [56]
Sung-Ching Lee 2017-3-26 3:49 PM
Formatted ... [58]
Sung-Ching Lee 2017-3-26 3:49 PM
Formatted ... [59]
Sung-Ching Lee 2017-3-26 3:49 PM
Formatted ... [60]
Nick 2017-3-8 1:54 PM
Formatted ... [61]
Nick 2017-3-1 3:01 PM

Sung-Ching Lee 2017-3-26 3:49 PM
Formatted ... [62]
Sung-Ching Lee 2017-3-26 3:49 PM
Formatted ... [63]
Sung-Ching Lee 2017-3-26 3:49 PM
Formatted ... [64]
Sung-Ching Lee 2017-3-26 3:49 PM
Formatted ... [65]
Sung-Ching Lee 2017-3-26 3:49 PM
Formatted ... [67]
Nick 2017-3-8 1:54 PM
Formatted ... [68]

Table 2: Comparison of annual NEE, $R_e$ and GEP, over different ecosystems (vegetation covers) in the Vancouver region using EC measurements. Sorted by magnitude of -NEE/GEP ratio.

| Site | Land cover | NEE | $R_e$ | GEP | -NEE/GEP |
|------|-----------|-----|-------|-----|----------|
| | | g C m$^{-2}$ year$^{-1}$ | | | |
| **Burns Bog** **(this study)** Delta, BC | Rewetted raised bog ecosystem | -179 | 236 | 415 | 43% |
| **Westham Island** (CA-Wes)[*] Delta, BC | Unmanaged grassland | -222 | 1215 | 1438 | 15% |
| **Campbell River** (CA-Ca1)[*] Vancouver Island | Douglas-fir forest (~55 yrs) | -328[+] | 1830[+] | 2158[+] | 15% |
| **Buckley Bay** (CA-Ca3)[*] Vancouver Island | Douglas-fir forest (~15 yrs) | 64[+] | 1487[+] | 1423[+] | -4% |

[*] Site identifier in global FLUXNET database (http://fluxnet.ornl.gov).
[+] Data from Krishnan et al., 2009 before fertilisation.

 **MINOR REVISION:**

**1) Line 149. "Longer gaps in $CO_2$ and $CH_4$ fluxes were filled …"**

[Response]

Thanks for the correction. We have changed the text to "Longer gaps in $CO_2$ and $CH_4$ fluxes were filled …".

1060

**2) Line 155. Valid data for EC-2 being for just 32% of the year. This needs more careful explanation and consideration. Later on it is implied that most of the missing data were for winter periods. Please provide a very brief summary of missing data by season. How does this affect the annual estimate? See also comment on Fig. 7.**

[Response]

1065   The information on seasonal data coverage has been added as requested:

"Valid data from EC-1 were obtained for 59% of the year (after quality control). Valid data from EC-2, which were restricted by power availability, was 32% of the year (after quality control). Data availability was the lowest in winter (38%/4% in winter, 71%/6% in spring, 67%/70% in summer, 60%/51% in fall, for EC-1/EC-2, respectively)."

1070   **3) Line 159. $CO_2$, not $C0_2$. Replace "(e.g." with "(i.e."**

[Response]

The font has been corrected and "e.g." has been changed to "i.e.".

**4) Line 161 "(GEP)" is unnecessary here. Suggest delete.**

1075   [Response]

Done. This has been deleted.

**5) Line 168. "r2 decides…" Replace "decides" with "determines".**

[Response]

1080   As suggested, "decides" has been replaced by "determines".

**6) Line 206. What approach was taken for gap filling $CH_4$ fluxes (or constraints on this approach) for parts of the year with large amounts of missing fluxes? See comments for Line 155, where the distribution of missing flux data not fully described, but implied that winter fluxes largely absent.**

1085   [Response]

Although there were only few values at low temperatures, they were consistently within the range of 5 to 9 nmol m$^{-2}$ s$^{-1}$, and were all included in parameterizing Equation 3. Due to the large gaps in winter, the smallest window size to provide reasonable results was 210 days, which means the model always has a reasonable and plausible fit for each day.

**7) Line 221. Missing word "effect of all long-lived …".**

[Response]

Done. We have changed the text to "The combined effect of all long-lived …".

**8) Section 4.3.1. Please include uncertainties for annual flux values in this text, or refer to relevant table.**

[Response]

The uncertainties for NEE (Line 266), GEP (Line 269) and $R_e$ (Line 269) have been added.

**9) Line 275. "early in the growing season".**

[Response]

Thanks, we have changed the text to "… early in the growing season …".

**10) Line 296. Cited annual NEE of -804 g C m$^{-2}$. This is a highly unlikely annual value. I do not have the time to check on all these references but suspect that there will be issues with methods or extrapolation to annual values. Could the authors please ensure that they consistently use the literature, e.g. don't mix chamber and EC studies, or at least note where very low confidence exists in some of these values.**

[Response]

The value of -804 g C m$^{-2}$ year$^{-1}$ was found in Anderson et al. (2016), but the measurements were not full-year measurements, and hence we agree that this value is not comparable. The text has been updated as follows:

"Other measurements, however, showed that restored wetlands were C sinks, all of them stronger than in this study, with NEE values ranging from -446 g C m$^{-2}$ year$^{-1}$ to -270 g C m-2 year$^{-1}$ (Badiou et al., 2011; Hendriks et al., 2007; Herbst et al., 2013; Knox et al., 2015)."

**11) Line 303. Please remove quote marks around Mer Bleue. This is a proper noun.**

[Response]

The quote mark has been removed.

**12) Line 304. Add "… than Burns Bog" or similar at the end of this sentence.**

[Response]

We have changed the text to "…both of which experienced a lower mean annual temperature than Burns Bog.".

**13) Line 305. 'dissolved organic carbon (DOC) export …"**

[Response]

We have changed the text to "It is important to estimate dissolved organic carbon (DOC) export to determine …".

1125 **14) Line 309. The headspace equilibration technique is not used to estimate DOC concentration. Please read the AGU poster abstract (that three of the authors of this paper are listed on as co-authors!) for the method used.**
[Response]
We appreciate the reviewer pointing out the error. The headspace equilibrium technique was only used for the determination of gases ($CO_2$, $CH_4$, $N_2O$) by D'Acunha et al. (2016) but not DOC. We have corrected the details regarding DOC analysis

1130 and DOC flux estimations. The text has been edited as follows:

"Estimation of DOC fluxes was based on regular (approx. monthly) water samples collected at 5 locations within the flux tower footprint. Water samples were analyzed for DOC concentrations using a TOC analyzer (Model TOC-VCSH, Shimadzu Scientific, Kyoto, Japan). Lateral water export was estimated as the residual of the water balance. D'Acunha et al.

1135 (2016) estimated DOC export for the current study area for Jan – Dec 2016 to be 22.4 g C $m^{-2}$ $year^{-1}$ (15% of annual NEE)."

**15) Line 309. Lateral flow? Suggest change to "lateral water export".**
[Response]
As suggested, we have changed the text to "Lateral water export was estimated as the residual of the water balance.".

1140

**16) Line 372. Goodrich et al. (2015) were looking for a mechanism for elevated CH4 fluxes occurring after rainfall events at a certain time of year, when the water table was within a narrow depth range. I don't think rain events can be cited as a suppression cause based on this reference. Please check carefully.**
[Response]

1145 We appreciate the comment. After consideration, we have decided to delete this statement and reference to avoid any ambiguity.

**17) Line 380-384. What about the annual flux reported by Goodrich et al. (2015)? 29 and 21 gCH$_4$-C $m^{-2}$ $yr^{-1}$ (EC, not chamber).**

1150 [Response]
This reference has been included and the text has been edited as follows:
"The $CH_4$ fluxes from a number of temperate and tropical pristine wetlands exceeded the $CH_4$ fluxes reported in this study, including emissions from marshes in the Southwestern US (130 g $CH_4$-C $m^{-2}$ $year^{-1}$, Whiting & Chanton, 2001), tropical wetlands in Costa Rica (82 g $CH_4$-C $m^{-2}$ $year^{-1}$, Nahlik & Mitsch, 2010), marshes in the Midwestern US (50 g $CH_4$-C $m^{-2}$

1155 $year^{-1}$, Koh et al., 2009), all three studies based on chamber measurements, and an ombrotrophic bog in New Zealand (29

and 21 g $CH_4$-C m$^{-2}$ year$^{-1}$ based on EC measurements, Goodrich et al., 2015). "

**18) Line 385 and Figure 7. I suggest that this figure be modified to show just summertime diurnal variation because:**
**1) annually-composed ensemble can mask season-specific diurnal variations; 2) apparently much of the wintertime data were missing (more needs to be said about this as noted above).**

[Response]

We appreciate reviewer's suggestion, the figure has been re-drawn.

[Figure]

The caption and the text have also been modified accordingly.

**19) Section 4.5 title. Suggest replace "exchange" with "balance".**

[Response]

We have changed the title to "$CO_2$e balance".

**20) Line 410. Replace "'long-term" with "annual".**

[Response]

We have changed the text to "... site processes but the determination of the annual budget requires reliable annual measurements.

**21) Line 412. "e.g. sustained step-change ….". I don't understand what this means. Please clarify.**

[Response]

After careful thought, we have decided to delete this term to avoid any ambiguity.

**22) Line 413. Please replace "were proposed" with "have been proposed".**

[Response]

Thanks, we have changed the text to "… and alternative models have been proposed …".

**23) Line 425. "Annual CH4 emission was" (this was a single year study).**

[Response]

We have changed the text to "Annual $CH_4$ emission was …".

**24) Line 430. Why exclude DOC flux? If it is because DOC flux was not measured in the subject year, then consider providing two NECB values, one without and one with estimated DOC.**

[Response]

We have revised the text as follows:

"In terms of the C balance (excluding DOC fluxes), our results suggest that our study area in BBECA was a net C sink (-163 ± 26.2 g C m$^{-2}$ year$^{-1}$) during the 8$^{th}$ year following rewetting. Combining $CO_2$, $CH_4$ and DOC fluxes resulted in a net C balance of -141 ± 26.2 g C m$^{-2}$ year$^{-1}$."

**25) Lines 433-436. I find this quite unsatisfying. "So what"? What useful message are we to take from this? A deeper philosophical discussion earlier would have helped, that questioned IPCC-type approaches as applied to restoration of long-term C sink ecosystems such as peatlands. What is the point of concluding with these diametrically opposing GWP estimates?**

[Response]

We appreciate reviewer's comment. To avoid any ambiguity, we have edited this paragraph. We agree that including the 20-year time horizon GWP estimate is unnecessary so we have removed the last sentence (Line 435-436). We have also slightly reworded the previous sentence (Line 433-435) as follows:

"In terms of net climate forcing of the system related to $CO_2$ and $CH_4$ fluxes expressed by GWPs, our results show that the ecosystem was almost $CO_{2e}$ neutral (-22 ± 103.1 g CO2e m-2 year-1) over a 100-year time horizon".